# On the Mechanisms of Collaborative Learning in VAE Recommenders

**Tung-Long Vuong**
Amazon
`longvt@amazon.com`

**Julien Monteil**
Amazon
`jul@amazon.com`

**Hien Dang**
University of Texas at Austin
`hiendang@utexas.edu`

**Volodymyr Vaskovych**
Amazon
`vaskovyc@amazon.com`

**Trung le**
Monash University
`trunglm@monash.com`

**Vu Nguyen**
Amazon
`vutngn@amazon.com`

## Abstract

Variational Autoencoders (VAEs) are a powerful alternative to matrix factorization for recommendation. A common technique in VAE-based collaborative filtering (CF) consists in applying binary input masking to user interaction vectors, which improves performance but remains underexplored theoretically. In this work, we analyze how collaboration arises in VAE-based CF and show it is governed by *latent proximity*: we derive a latent sharing radius that informs when an SGD update on one user strictly reduces the loss on another user, with influence decaying as the latent Wasserstein distance increases. We further study the induced geometry: with clean inputs, VAE-based CF primarily exploits *local* collaboration between input-similar users and under-utilizes *global* collaboration between far-but-related users. We compare two mechanisms that encourage *global* mixing and characterize their trade-offs: ① $\beta$-KL regularization directly tightens the information bottleneck, promoting posterior overlap but risking representational collapse if too large; ② input masking induces stochastic *geometric* contractions and expansions, which can bring distant users onto the same latent neighborhood but also introduce neighborhood drift. To preserve user identity while enabling global consistency, we propose an anchor regularizer that aligns user posteriors with item embeddings, stabilizing users under masking and facilitating signal sharing across related items. Our analyses are validated on the Netflix, MovieLens-20M, and Million Song datasets. We also successfully deployed our proposed algorithm on an Amazon streaming platform following a successful online experiment.

## 1 Introduction

Recommender systems are essential for delivering personalized user experiences. Having benefited from three decades of research (Resnick et al., 1994), Collaborative filtering (CF) remains an essential approach in today's recommender systems (Zhu et al., 2025). CF predictive process consists in predicting user preferences by identifying and leveraging similarity patterns between users and items (Su & Khoshgoftaar, 2009; Ricci et al., 2010), which naturally aligns with the framework of latent variable models (LVMs) (Bishop & Nasrabadi, 2006), where latent representations are used to capture the shared structure of user-item interactions. Due to their simplicity and effectiveness, LVMs have historically played a central role in CF research. However, these models are inherently linear, which limits their capacity to model the complex and non-linear nature of real-world user behavior (Paterek, 2007; Mnih & Salakhutdinov, 2007). To overcome these limitations, researchers have increasingly explored the integration of neural networks (NNs) into CF frameworks, enabling more expressive modeling and yielding notable improvements in recommendation accuracy (He et al., 2017; Wu et al., 2016; Liang et al., 2018; Truong et al., 2021; Li et al., 2021). A particularly successful line of work is VAE-based collaborative filtering (Liang et al., 2018), which extends the variational autoencoder (VAE) framework (Kingma & Welling, 2013; Rezende et al., 2014) to collaborative filtering tasks. Unlike traditional latent factor models (Hu et al., 2008; Paterek, 2007; Mnih & Salakhutdinov,

2007), which require learning a separate latent vector for each user, VAE-based models offer a user-independent parameterization, where the number of trainable parameters remains fixed regardless of the number of users Lobel et al. (2019), leading to remarkable scalability. Additionally, empirically, VAE-based CF models consistently outperform many existing LVM-based alternatives (Liang et al., 2018; Kim & Suh, 2019; Walker et al., 2022; Ma et al., 2019; Guo et al., 2022; Wang et al., 2023; 2022; Guo et al., 2024; Husain & Monteil, 2024; Li et al., 2021; Tran & Lauw, 2025).

A key driver of the strong performance of VAE-based CF is the use of a binary mask that corrupts the user interaction vector, producing a partial history from which the model is trained to reconstruct the full interaction vector (Liang et al., 2018). This masking strategy has been shown to significantly enhance recommendation accuracy (see Section 4.2 and Figure 2, comparing settings with and without masking). Although input noise has become a standard component in VAE-based CF models due to its empirical effectiveness, its underlying mechanisms and potential side effects remain largely unexplored. Existing works treat masking as a simple performance-enhancing heuristic, without thoroughly examining how it influences the learning process or affects the latent representations. Our work aims to fill this gap with a comprehensive study the effect of input noise in VAE-based CF. The contributions of this paper are summarized as follows:

- We conduct an in-depth analysis of the collaboration mechanism in VAE-based collaborative filtering models and reveal that ① collaboration in VAE-CF is fundamentally governed by latent proximity; ② VAE-CF with clean inputs primarily leverages local collaboration and fails to utilize global collaborative signals when input distances are large; ③ Both $\beta$-KL regularization and input masking can encourage global collaborative signals, but they operate through distinct mechanisms with different trade-offs i.e., *$\beta$-KL regularization* promotes posterior mixing by directly constraining the information bottleneck, but suffers the risk of representational collapse when applied too aggressively while *Input masking* achieves mixing through *geometric* and stochastic means such that it can bring distant users into the same latent neighborhood, and latent space expansions can introduce neighborhood drift effects.

- Guided by our theoretical analysis, we propose a regularization scheme that addresses the issues induced by input masking, mitigating the loss of local collaboration while preserving its benefits for global alignment. Specifically, we model items as learnable anchors in latent space, and during training, the masked encoder outputs are pulled toward the user's anchor centroid. This acts as a training-only auxiliary condition that helps stabilize user representations under input corruption without tightening the information bottleneck, promoting consistent, semantically grounded latent proximity. *To our knowledge, ours is the first work to systematically analyze the collaboration mechanisms in VAE-based CF, showing that both $\beta$-KL regularization and input masking can promote* global *collaboration. In contrast to prior works that focus on addressing the problems of $\beta$-KL regularization (ref. Appendix B for a detailed discussion), we address the issues induced by input masking.*

- Our experimental results demonstrate the strong benefits of the proposed PIA approach compared to vanilla VAE-CF on benchmark datasets and especially the success on the A/B testing at Amazon streaming platform. We conducted ablation studies across user groups segmented by interaction count to validate the effectiveness of global collaborative signals. Additionally, we provide visualizations of the learned latent space that support our theoretical analysis.

## 2 COLLABORATION MECHANISM IN VAE-BASED CF

**Notations.** We index users by $u \in \{1, 2, \ldots, U\}$ and items by $i \in \{1, 2, \ldots, I\}$, where $U, I$ are the number of users and items, respectively. $\mathbf{X} \in \{0, 1\}^{U \times I}$ represents the user-item interaction matrix (e.g., click, watch, check-in, etc.) and $\mathbf{x}_u = [\mathbf{x}_{u1}, \mathbf{x}_{u2}, \ldots, \mathbf{x}_{uI}]$ is an $I$-dimensional binary vector (the $u$-th row of $\mathbf{X}$) whereby $\mathbf{x}_{ui} = 1$ implies that user $u$ has interacted with item $i$ and $\mathbf{x}_{ui} = 0$ indicates otherwise. Note that $\mathbf{x}_{ui} = 0$ does not necessarily mean user $u$ dislikes item $i$; the item may never be shown to the user. For simplicity, we use $\mathbf{x}$ to denote a general user interaction vector and retain $\mathbf{x}_u$ when specifically referring to user $u$. Additionally, we measure distances between input vectors with the $\ell_1$ norm $\| \cdot \|_1$ (Hamming) and between latent distributions with the 1-Wasserstein distance $W_1(\cdot, \cdot)$. We refer to the Appendix Table A for the notation summary.

## 2.1 VAE-BASED COLLABORATIVE FILTERING

Given a user's interaction history $\mathbf{x} = [\mathbf{x}_1, \mathbf{x}_2, \ldots, \mathbf{x}_I]^\top \in \{0,1\}^I$, the goal is to predict the full interaction behavior of this user with all remaining items. To simulate this process during training and to avoid overfitting to non-informative patterns (Steck, 2020), *dropout-style random masking is commonly used* in VAE-based CF (Wu et al., 2016; Liang et al., 2018; Lobel et al., 2019; Shenbin et al., 2020; Vančura & Kordík, 2021) as a form of stochastic input corruption. Concretely, we introduce a random binary mask $\mathbf{b} \in \{0,1\}^I$ is , with the entry 1 as *un-masked*, and 0 as *masked*. Thus, $\mathbf{x}_h = \mathbf{x} \odot \mathbf{b}$ is the user's partial interaction history, the goal is to recover the full $\mathbf{x}$ given $\mathbf{x}_h$.

**Training Objective.** The parameters $\phi, \theta$ of the VAE-based collaborative filtering model are learnt by minimizing the negative $\beta$-regularized Evidence Lower Bound (ELBO):

$$\mathcal{L}_{\text{VAE}}(\mathbf{x}; \theta, \phi) = -\mathbb{E}_{q_\phi(\mathbf{z}|\mathbf{x}_h)}\big[\log p_\theta(\mathbf{x} \mid \mathbf{z})\big] + \beta \, \text{KL}\big(q_\phi(\mathbf{z} \mid \mathbf{x}_h) \,\|\, p(\mathbf{z})\big), \tag{1}$$

with standard reparameterization

$$\mathbf{b} \sim \text{Bernoulli}(\rho)^I, \quad \mathbf{x}_h = \mathbf{x} \odot \mathbf{b}, \quad (\boldsymbol{\mu}, \boldsymbol{\sigma}^2) = q_\phi(\mathbf{x}_h), \ \boldsymbol{\epsilon} \sim \mathcal{N}(\mathbf{0}, \mathbf{I}), \ \mathbf{z} = \boldsymbol{\mu} + \boldsymbol{\epsilon}\boldsymbol{\sigma} \tag{2}$$

where $\boldsymbol{\rho}$ is the hyperparameter of a Bernoulli distribution, $q_\phi$ is a $\phi$-parameterized neural network, which outputs the mean $\boldsymbol{\mu}$ and variance $\boldsymbol{\sigma}^2$ of the Gaussian distribution.

## 2.2 COLLABORATIVE LEARNING VIA VAE-CF

Before presenting our theoretical analysis, we briefly recall the goal of collaborative filtering (CF): to predict a user's preferences from *other users' interaction patterns*, without relying on user/item side information. Figure 1 highlights two common forms of cross-user information in CF.

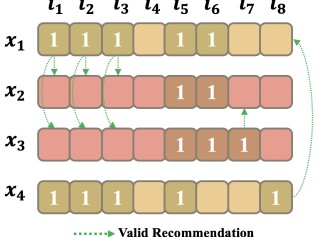

**(1) Neighborhood transfer.** Users $\mathbf{x}_1$ and $\mathbf{x}_4$ are close in the input space (e.g., under $\ell_1$), so each serves as a natural reference for the other: similar interaction histories should lead to similar predictions.

**(2) Far-but-related transfer.** Users can be far in $\ell_1$ yet still meaningfully related through shared positives. Let $S_{\mathbf{x}_u} = \{i : \mathbf{x}_{ui} = 1\}$ denote the set of positive items for user $u$. If $S_{\mathbf{x}_2} \subset S_{\mathbf{x}_1}$, then

Figure 1: *Local* and *global* collaborative signal example.

$\mathbf{x}_1$ is a more active user with similar interests and can inform recommendations for $\mathbf{x}_2$ (e.g., items $i_1, i_2, i_3 \in S_{\mathbf{x}_1} \setminus S_{\mathbf{x}_2}$), even though $\mathbf{x}_1$ and $\mathbf{x}_2$ need not be close in $\ell_1$. We refer to this "far-but-related" influence as *Global collaborative signal*.

To formalize these notions, for $\delta > 0$ we define the input-space neighborhood of user $u$ as

$$N_\delta(u) = \{ v : \|\mathbf{x}_u - \mathbf{x}_v\|_1 \le \delta \},$$

and the (nonzero) overlap indicator as

$$\text{ov}(\mathbf{x}_u, \mathbf{x}_v) = \mathbf{1}_{\left\{ \left| S_{\mathbf{x}_u} \cap S_{\mathbf{x}_v} \right| > 0 \right\}}.$$

**Definition 2.1** (Local collaborative signal). The prediction for user $u$ depends on other users within the neighborhood $N_\delta(u)$.

**Definition 2.2** (Global collaborative signal). A model exhibits a *global* collaborative signal at scale $\delta$ if there exist users $u$ and $v$ such that $\|\mathbf{x}_u - \mathbf{x}_v\|_1 > \delta$ and $\text{ov}(\mathbf{x}_u, \mathbf{x}_v) = 1$, and their predictions strictly influence each other.

**Cross-user influence via gradient transfer.** In VAE-CF, cross-user information is exchanged primarily during *training* because the same parameters are updated using many different users. We therefore operationalize "influence" as *training-time transfer*: user $v$ influences user $u$ if an SGD step computed from $v$ decreases $u$'s expected loss to first order. The following theorem formalizes this notion and shows that such transfer is governed by proximity of the users' posteriors in latent space.

**Theorem 2.3** (Latent-$W_1$ sharing radius). *Assume decoder gradient is Lipschitz in $z$ such that for any given $\mathbf{z} \sim q_\phi(\cdot \mid \mathbf{x}_u)$, uniformly in $\mathbf{x}$ and $\forall \mathbf{z}'$, $\|\nabla_\theta \ell_\theta(\mathbf{x}, \mathbf{z}) - \nabla_\theta \ell_\theta(\mathbf{x}, \mathbf{z}')\| \le L_{\theta z} \|\mathbf{z} - \mathbf{z}'\|$. Let*

$$q_u := q_\phi(\cdot \mid \mathbf{x}_u), \quad q_v := q_\phi(\cdot \mid \mathbf{x}_v), \quad \mathcal{L}_u(\theta) := \mathbb{E}_{q_u}[\ell_\theta(\mathbf{x}_u, \mathbf{z})], \quad g_u(\theta) := \nabla_\theta \mathcal{L}_u(\theta).$$

*Define the content-mismatch term*

$$\Delta_x(u,v) := \Big\| \mathbb{E}_{q_v}\big[ \nabla_\theta \ell_\theta(\mathbf{x}_u, \mathbf{z}) - \nabla_\theta \ell_\theta(\mathbf{x}_v, \mathbf{z})\big] \Big\|, \quad D_{u,v} := L_{\theta z}\, W_1(q_u, q_v) + \Delta_x(u,v).$$

*For one SGD step on user $v$, $\theta^+ = \theta - \eta\, g_v(\theta)$, the first-order change satisfies*

$$\mathcal{L}_u(\theta^+) - \mathcal{L}_u(\theta) \;\leq\; -\eta\, \|g_u(\theta)\|\big(\, \|g_u(\theta)\| - D_{u,v}\,\big) \;+\; O(\eta^2).$$

*Consequently, the step on $v$ strictly decreases $\mathcal{L}_u$ to first order whenever $D_{u,v} < \|g_u(\theta)\|$, i.e.*

$$W_1(q_u, q_v) \;<\; r_{\text{share}}(u,v;\theta) \;:=\; \frac{[\|g_u(\theta)\| - \Delta_x(u,v)]_+}{L_{\theta z}}.$$

*Proof.* See Appendix E.1. □

**Result-1 (Collaboration is governed by *latent* proximity).** Theorem 2.3 characterizes when an SGD step on user $v$ is beneficial for user $u$. The key quantity is the *transfer penalty* $D_{u,v} = L_{\theta z}\, W_1(q_u, q_v) + \Delta_x(u,v)$, which has two components: (i) a **latent mismatch** term $W_1(q_u, q_v)$ measuring how close the users' posteriors are in latent space, and (ii) a **content mismatch** term $\Delta_x(u,v)$ capturing how different the decoder gradients are even when evaluated at the same latent code. To first order, a step on $v$ decreases $\mathcal{L}_u$ whenever $D_{u,v} < \|g_u(\theta)\|$. Therefore, gradient sharing is effectively restricted to a *latent neighborhood* around $u$, and the strength of collaboration decays as users become latently farther apart.

Consequently, Theorem 2.3 reduces the distinction between *local* and *global* collaborative signals (Definitions 2.1–2.2) to a geometric question: which user pairs become *latently close* enough to fall within the sharing radius. Local signals correspond to input-close pairs that remain latently close, whereas global signals require certain input-distant but overlap-related pairs to be mapped close in latent space. In the next section, we examine how the *geometry of the input space* (including masking) shapes the *geometry of the latent space* in VAE-based collaborative filtering.

## 2.3 Impact of the Geometry of the Input and Latent Spaces

We summarize the key theoretical results and present the discussion subsequently.

**Lemma 2.4.** *Assume the encoder is $L_\phi$-Lipschitz, i.e., $\|q_\phi(\mathbf{x}_u) - q_\phi(\mathbf{x}_v)\| \leq L_\phi \|\mathbf{x}_u - \mathbf{x}_v\|\ \forall \mathbf{x}_u, \mathbf{x}_v \in \{0,1\}^I$ then $W_1\big(q_\phi(\cdot \mid \mathbf{x}_u), q_\phi(\cdot \mid \mathbf{x}_v)\big) \leq L_\phi \|\mathbf{x}_v - \mathbf{x}_u\|_1, \forall \mathbf{x}_u, \mathbf{x}_v \in \{0,1\}^I.$*

**Theorem 2.5.** *Assume $p_\theta(\mathbf{x} \mid \mathbf{z})$ is a regular exponential family with sufficient statistics $T(\mathbf{x})$, natural parameter $\eta(\mathbf{z})$, and log-partition $A$. Let $q_u = q_\phi(\cdot \mid \mathbf{x}_u)$ and $q_v = q_\phi(\cdot \mid \mathbf{x}_v)$, and define $\alpha(\mathbf{z}) = \frac{q_u(\mathbf{z})}{q_u(\mathbf{z}) + q_v(\mathbf{z})}$. Then the $\beta$-regularized pairwise objective satisfies*

$$\min_{\eta(\cdot)} \left\{ \sum_{i \in \{u,v\}} \mathbb{E}_{q_i}\big[ -\log p_\theta(\mathbf{x}_i \mid \mathbf{z})\big] + \beta \sum_{i \in \{u,v\}} \mathrm{KL}\big(q_\phi(\mathbf{z} \mid \mathbf{x}_i) \,\|\, p(\mathbf{z})\big) \right\} \tag{3}$$
$$= C + \int \big(q_u(\mathbf{z}) + q_v(\mathbf{z})\big)\, \Delta_{A^*}\big(\mathbf{x}_u, \mathbf{x}_v; \alpha(\mathbf{z})\big)\, d\mathbf{z} + \beta \sum_{i \in \{u,v\}} \mathrm{KL}\big(q_\phi(\mathbf{z} \mid \mathbf{x}_i) \,\|\, p(\mathbf{z})\big),$$

*where $C$ is a constant, $A^*$ is the convex conjugate of $A$, and for $\alpha \in [0,1]$*

$$\Delta_{A^*}\big(\mathbf{x}_u, \mathbf{x}_v; \alpha\big) = \alpha\, A^*\big(T(\mathbf{x}_u)\big) + (1-\alpha)\, A^*\big(T(\mathbf{x}_v)\big) - A^*\big(\alpha T(\mathbf{x}_u) + (1-\alpha)T(\mathbf{x}_v)\big) \;\geq\; 0,$$

*with equality iff $T(\mathbf{x}_u) = T(\mathbf{x}_v)$.*

*Additionally, assume the prior $p$ satisfies the Bobkov–Götze/Talagrand $T_1(C)$ inequality (Bobkov & Götze, 1999) with a constant $C > 0$ (e.g., Normal prior $p = \mathcal{N}(0, \sigma^2 I)$):*

$$W_1\big(q_u, q_v\big) \;\leq\; \Big( \sqrt{2C\, \mathrm{KL}(q_u \| p)} \;+\; \sqrt{2C\, \mathrm{KL}(q_v \| p)} \Big). \tag{4}$$

*Proof.* See Appendix E.2. □

**Result-2 (Clean inputs favor local over global collaboration).**

- *Local neighborhoods are preserved.* Lemma 2.4 implies that $\ell_1$-near users induce nearby posteriors under $W_1$ (up to the encoder Lipschitz constant), so input-space locality naturally maps to latent-space locality, preserving local collaboration.

- *Reconstruction discourages overlap for mismatched users.* Theorem 2.5 shows that when $T(\mathbf{x}_u) \neq T(\mathbf{x}_v)$, overlapping posteriors incur a strictly positive "compromise gap" $\int (q_u + q_v) \Delta_{A^*} d\mathbf{z}$ in Eq. (3). This pushes the model to separate $q_u$ and $q_v$, making it difficult for input-distant, content-mismatched users to become latent-close, discouraging global collaboration.

- *$\beta$-KL encourages posterior overlap.* Increasing $\beta$ reduces the optimal KL terms, tightening the upper bound of $W_1(q_u, q_v)$, thereby increasing latent overlap i.e., potentially bringing input-distant users closer in latent space and enabling global collaboration. However, overly strong KL pressure risks posterior collapse and weakens recommendation quality; in practice $\beta$ is typically kept small in ranking-focused CF.

Putting these pieces together, with clean (unmasked) inputs and moderate $\beta$, similar users remain latent-close while content-mismatched users are pushed apart, yielding a clustered latent geometry that mirror input similarity (Setting-1, Figure 2). By Theorem 2.3, SGD updates are therefore shared predominantly *within* these latent neighborhoods (i.e., across pairs with $W_1(q_u, q_v) < r_{\text{share}}(u, v; \theta)$). In other words, under clean inputs, VAE-CF primarily exploits *local* collaborative signals and suppresses global influence from input-distant users.

**Theorem 2.6.** *Let $\mathbf{x}_u, \mathbf{x}_v \in \{0,1\}^I$ be binary inputs. Let $b_{\mathbf{x}_u}, b_{\mathbf{x}_v} \sim \mathrm{Bern}(\rho)^I$ be independent masks and set $\mathbf{x}'_u = \mathbf{x}_u \odot b_{\mathbf{x}_u}$ and $\mathbf{x}'_v = \mathbf{x}_v \odot b_{\mathbf{x}_v}$. Denote the number of non-overlapped items $h = \|\mathbf{x}_u - \mathbf{x}_v\|_1$ and the number of overlapped items $s = \langle \mathbf{x}_u, \mathbf{x}_v \rangle$. For any $\delta > 0$, define $T_\delta = \lceil \delta \rceil - 1$ and $U_\delta = \lceil \delta \rceil$. Then:*

***Contraction.*** $\Pr\big[\|\mathbf{x}'_u - \mathbf{x}'_v\|_1 < \delta\big] \geq \big(\rho^2 + (1-\rho)^2\big)^s \sum_{k=0}^{\min\{h, T_\delta\}} \binom{h}{k} \rho^k (1-\rho)^{h-k}.$

***Expansion.*** $\Pr\big[\|\mathbf{x}'_u - \mathbf{x}'_v\|_1 \geq \delta\big] \geq \sum_{k=U_\delta}^{s} \binom{s}{k} \big(2\rho(1-\rho)\big)^k \big(1 - 2\rho(1-\rho)\big)^{s-k}.$

*Proof.* See Appendix E.3. $\qquad\square$

**Result-3 (Masking induces *stochastic* neighborhood mixing).** Theorem 2.6 quantifies how random masking can *stochastically* contract or expand the $\ell_1$ distance between two users' interaction vectors. Under Lemma 2.4, contractions propagate to latent space; when the realized distance falls within the sharing radius (Theorem 2.3), an SGD step on $v$ benefits $u$, even if the original (unmasked) pair was far apart. This creates *intermittent* long-range sharing events that inject global collaborative signal. Conversely, expansions can push genuine neighbors outside the sharing radius, increasing gradient variance and weakening local transfer. The net effect is global mixing through occasional contractions at the cost of neighborhood drift from expansions (Setting-2, Figure 2).

## 2.4 BEYOND LATENT GEOMETRY: $\beta$–KL VS. MASKING

The previous section showed that collaboration is governed by *latent* proximity via the sharing radius (Theorem 2.3). Geometrically, both strengthening the KL penalty and introducing masking reduce latent distances and can enable global collaboration. We now turn these geometric insights into *actionable* guidance by linking latent distances to the KL terms optimized in the ELBO.

Recall the VAE-CF setting in Eq (1 & 2) with masked inputs $\mathbf{x}_h = \mathbf{x} \odot \mathbf{b}$, where $\mathbf{b} \sim \mathrm{Bern}(\rho)^I$ i.i.d. across items and $\rho=1$ recovers the clean-input setting.

For any users $u, v$ and masks $\mathbf{b}_u, \mathbf{b}_v$, Theorem 2.5 (transportation inequality) yields

$$W_1\big(q_\phi(\cdot \mid \mathbf{x}_u \odot \mathbf{b}_u), q_\phi(\cdot \mid \mathbf{x}_v \odot \mathbf{b}_v)\big) \leq \sqrt{2C \, \mathrm{KL}\big(q_\phi(\cdot \mid \mathbf{x}_u \odot \mathbf{b}_u) \| p\big)} + \sqrt{2C \, \mathrm{KL}\big(q_\phi(\cdot \mid \mathbf{x}_v \odot \mathbf{b}_v) \| p\big)}.$$
(5)

Averaging uniformly over user pairs $(u, v)$ and independently sampled masks $\mathbf{b}_u, \mathbf{b}_v$ and using Jensen's inequality (concavity of the square root) gives

$$\mathbb{E}_{u,v,\mathbf{b}} \, W_1\big(q_\phi(\cdot \mid \mathbf{x}_u \odot \mathbf{b}_u), q_\phi(\cdot \mid \mathbf{x}_v \odot \mathbf{b}_v)\big) \leq 2\sqrt{2C} \, \sqrt{\mathbb{E}_{\mathbf{x},\mathbf{b}} \, \mathrm{KL}\big(q_\phi(\cdot \mid \mathbf{x} \odot \mathbf{b}) \| p\big)}. \quad (6)$$

Define the aggregated (masked) posterior $q_h(Z) := \mathbb{E}_{X,B}\, q_\phi(Z \mid X \odot B)$ [1], and the encoder mutual information $I_{q_\phi}(X_h; Z) := \mathbb{E}_{\mathbf{x},\mathbf{b}}\, \mathrm{KL}\big(q_\phi(Z \mid \mathbf{x} \odot \mathbf{b}) \,\|\, q_h(Z)\big)$, a standard identity then gives

$$\mathbb{E}_{\mathbf{x},\mathbf{b}}\, \mathrm{KL}\big(q_\phi(Z \mid \mathbf{x} \odot \mathbf{b}) \,\|\, p(Z)\big) \;=\; I_{q_\phi}(X_h; Z) \;+\; \mathrm{KL}\big(q_h(Z) \,\|\, p(Z)\big). \tag{7}$$

Combining 6 and 7, any mechanism that lowers $I_{q_\phi}(X_h; Z)$ and/or $\mathrm{KL}(q_h\|p)$ *shrinks* expected pairwise latent distances, increasing the probability that user pairs fall within the sharing radius (Theorem 2.3). This leads to two directions for encouraging global collaboration:

$\beta$-**KL (objective-level).** Increasing $\beta$ in the training objective (Eq. 1) strengthens the KL penalty, and by the decomposition in Eq. (7) this effectively penalizes the sum $I_{q_\phi}(X_h; Z) + \mathrm{KL}(q_h\|p)$. At convergence this typically drives both terms down, which in turn induces a near-uniform contraction of inter-user latent distances and shrinks the right-hand side of Eq. (6). The trade-off is that overly large $\beta$ weakens the reconstruction signal and risks posterior collapse, degrading user semantics. A common solution to alleviate collapse is to use a more expressive prior (often multi-modal, e.g., mixtures, VampPrior, or flow-based priors). However, an expressive prior also weakens the global pairwise contraction effect compared to a simple prior such as the standard normal prior. Specifically, with a Gaussian prior, the KL term pulls all user posteriors toward a common zero-centered basin, so user pairs have a higher chance of becoming latently close, strengthening sharing at a fixed $\beta$. In contrast, expressive priors can satisfy the KL by distributing users across different modes rather than contracting them to a single center, making collaboration locally dependent on mode assignment.

**Input masking (data-level).** Smaller values of $\rho$ (Eq. 2) increase the masking strength, reducing the information available in $X_h$. By the data-processing inequality, this tends to decrease $I_{q_\phi}(X_h; Z)$ and often also the aggregate mismatch $\mathrm{KL}(q_h\|p)$, tightening the bound in Eq. (6). However, masking affects the geometry of the latent space in a more stochastic manner than the uniform contraction produced by increasing $\beta$. Random masks cause sample-dependent contractions and expansions across batches: sometimes two users who are far apart in input space become close in latent space, promoting desirable long-range sharing, while in other cases genuinely similar users may be pushed apart, weakening local reliability. For an individual user, each new mask realization slightly shifts the latent posterior, so the user's nearest-neighbor set fluctuates from batch to batch. Over training, these shifts accumulate, producing what we refer to as *neighborhood drift*: the local structure of the user's neighborhood wanders, and the gradients shared among nearby users become noisy and inconsistent.

## 3 PROPOSED METHOD: PERSONALIZED ITEM ALIGNMENT (PIA)

Prior work has primarily pursued the objective-level pathway (adjusting $\beta$ or redesigning $p(\mathbf{z})$; see Section B) while largely treating input masking as a benign training trick. In this section, we explore an alternative direction for improving VAE-CF: addressing the downside of masking by stabilizing the stochastic geometry it induces, so that beneficial long-range sharing occurs more consistently and with reduced drift. We introduce a *training-only* regularizer that pushes the masked posterior $q_\phi(\mathbf{z} \mid \mathbf{x}_h)$ toward a user-specific target constructed from the user's positive items. This stabilizes the masking-induced geometry without changing test-time inference, ensuring that different masked views of the same user map to a consistent latent region while bringing users with overlapping items closer in a semantically grounded way. We define the overall objective as:

$$\mathcal{L}_{\text{PIA-VAE}}(\mathbf{x}; \theta, \phi, E) \;=\; \mathbb{E}_{\mathbf{b}}\Big[\mathcal{L}_{\text{VAE}}(\mathbf{x}; \theta, \phi; \mathbf{x}_h)\Big] \;+\; \lambda_A\, \mathbb{E}_{\mathbf{b}}\Big[\mathcal{L}_A(\mathbf{x}_h, \mathbf{x}; \phi, E)\Big], \tag{8}$$

where $E = \{\mathbf{e}_i \in \mathbb{R}^d\}_{i=1}^I$ are learnable *item anchors* in latent space (same dimension as $\mathbf{z}$), and $\lambda_A > 0$ is small. In particular, let $S_{\mathbf{x}} = \{i : \mathbf{x}_i = 1\}$ be the positives for user $\mathbf{x}$, we have:

$$\mathcal{L}_A(\mathbf{x}_h, \mathbf{x}; \phi, E) \;=\; \frac{1}{|S_{\mathbf{x}}|} \sum_{i \in S_{\mathbf{x}}} \mathbb{E}_{\mathbf{z} \sim q_\phi(\mathbf{z}|\mathbf{x}_h)}\big[\, \|\mathbf{z} - \mathbf{e}_i\|_2^2 \,\big]. \tag{9}$$

---

[1]We use uppercase letters for random variables e.g., $X$ and bold lowercase for their realizations $\mathbf{x}$.

**Proposition 3.1.** *Assume the encoder posterior is diagonal-Gaussian,* $q_\phi(\mathbf{z} \mid \mathbf{x}_h) = \mathcal{N}(\boldsymbol{\mu}_\phi(\mathbf{x}_h), \mathrm{diag}(\boldsymbol{\sigma}_\phi^2(\mathbf{x}_h)))$. *Let the item centroid be* $\bar{\mathbf{e}}_\mathbf{x} := \frac{1}{|S_\mathbf{x}|} \sum_{i \in S_\mathbf{x}} \mathbf{e}_i$, *then*

$$\mathcal{L}_\mathrm{A}(\mathbf{x}_h, \mathbf{x}; \phi, E) = \underbrace{\left\| \boldsymbol{\mu}_\phi(\mathbf{x}_h) - \bar{\mathbf{e}}_\mathbf{x} \right\|_2^2}_{\text{align mean to item centroid}} + \underbrace{\mathrm{tr}\, \Sigma_\phi(\mathbf{x}_h)}_{\text{variance shrinkage}} + const(\mathbf{x}, E), \tag{10}$$

*where* $\Sigma_\phi(\mathbf{x}_h) = \mathrm{diag}(\boldsymbol{\sigma}_\phi^2(\mathbf{x}_h))$ *and* $const(\mathbf{x}, E) = \frac{1}{|S_\mathbf{x}|} \sum_{i \in S_\mathbf{x}} \|\mathbf{e}_i\|_2^2 - \|\bar{\mathbf{e}}_\mathbf{x}\|_2^2$.

*Proof.* See Appendix E.4. □

Proposition 3.1 indicates that PIA ① centers masked latents near the user's *item barycenter* $\bar{\mathbf{e}}_\mathbf{x}$ and ② modestly reduces posterior spread. Two users $u, v$ with similar positive-item sets have close centroids, so their masked posteriors become *latently close* more frequently, increasing the chance they fall within the sharing radius and benefit from each other's updates.

**Proposition 3.2.** *Fix* $\mathbf{x}$ *and its neighborhood in which the ELBO objective* $\mathcal{L}_{VAE}(\mathbf{x}; \theta, \phi; \mathbf{x}_h)$, *defined in Eq. (1), written as a function of the encoder mean* $\boldsymbol{\mu}_\phi(\mathbf{x}_h)$, *admits a quadratic approximation with Hessian* $H \succeq mI$ *and* $\|H\| \leq L$. *Adding* $\lambda_\mathrm{A} \|\boldsymbol{\mu}_\phi(\mathbf{x}_h) - \bar{\mathbf{e}}_\mathbf{x}\|_2^2$ *to this objective yields an effective Hessian* $H_{eff} = H + 2\lambda_\mathrm{A} I$. *Let* $\boldsymbol{\mu}^{(0)}(\mathbf{x}_h)$ *be the unregularized minimizer over masks and* $\boldsymbol{\mu}^{(A)}(\mathbf{x}_h)$ *the minimizer with alignment. Then with* $\tau = \left(\frac{L}{L + 2\lambda_\mathrm{A}}\right)$, *we obtain the following inequalities*

$$\mathrm{Var}_\mathbf{b}\big[\boldsymbol{\mu}^{(A)}(\mathbf{x}_h)\big] \preceq \tau^2 \mathrm{Var}_\mathbf{b}\big[\boldsymbol{\mu}^{(0)}(\mathbf{x}_h)\big], \qquad \mathbb{E}_\mathbf{b}\big[\|\boldsymbol{\mu}^{(A)}(\mathbf{x}_h) - \bar{\mathbf{e}}_\mathbf{x}\|^2\big] \leq \tau \mathbb{E}_\mathbf{b}\big[\|\boldsymbol{\mu}^{(0)}(\mathbf{x}_h) - \bar{\mathbf{e}}_\mathbf{x}\|^2\big].$$

*Proof.* See Appendix E.5. □

Proposition 3.2 indicates that adding the PIA term makes the masked encoder *locally better conditioned* and pulls its mean $\mu_\phi(\mathbf{x}_h)$ toward a per-user item centroid. Quantitatively, it shrinks ① the variance of $\mu_\phi(\mathbf{x}_h)$ across different masks and ② the average drift of $\mu_\phi(\mathbf{x}_h)$ from the centroid by a multiplicative factor $\tau \in (0, 1)$. Hence, **masked views of the same user are more alike and less noisy**, so the neighborhoods we train on are closer to the neighborhoods we infer on at test time.

In summary, PIA ① *stabilizes the geometric pathway:* aligning $q_\phi(\mathbf{z} \mid \mathbf{x}_h)$ to a fixed per-user $\bar{\mathbf{e}}_\mathbf{x}$ reduces masked-vs-clean drift and gradient variance; expansions are less likely to eject genuine neighbors from the sharing radius; ② *promotes meaningful global mixing:* shared items pull users toward nearby centroids, creating consistent, semantically grounded latent proximity instead of relying purely on stochastic contractions ((Setting-3, Figure 2)); ③ *introduces no test-time burden:* $E$ and the regularizer are estimated during training-only; inference uses the standard $q_\phi(\mathbf{z} \mid \mathbf{x})$.

**Remark: geometric view of collaborative filtering.** The anchors $\mathbf{e}_i$ form a semantic map where co-liked items organize into neighborhoods, and $\bar{\mathbf{e}}_\mathbf{x}$ lies within the convex hull of the user's liked items. This implies:

- **Users with large/diverse interaction histories.** When $|S_\mathbf{x}|$ is large and diverse, $\bar{\mathbf{e}}_\mathbf{x}$ may be less peaked; PIA then primarily improves *stability* rather than enforcing a specific prototype. Crucially, users who share more items (or items from the same neighborhoods) have *closer centroids*, making their representations neighbors and strengthening collaborative signal via the sharing radius (Theorem 2.3). For multi-modal users, the barycenter lies between item neighborhoods, keeping them close to multiple communities; shared (or nearby) items still pull such users into overlapping regions, allowing global signals to flow from both sides. In effect, PIA brings similar users closer without collapsing diverse preferences into a single artificial mode.

- **Cold-start users.** Warm users sculpt the semantic map: items they co-like are pulled into coherent neighborhoods. Cold users, even with few interactions, are placed into this *already-organized* map by aligning toward their clicked items' anchors. Each positive item is a landmark surrounded by warm users who also liked it (and related items). Aligning cold users toward these landmarks means that even a single shared item can snap them into warm clusters. Moreover, because items sharing users are pulled closer under PIA, interacting with a similar item also places cold users in the *right* neighborhood. In click space, such users appear far from any warm user; in latent space, one or two well-placed items act as shortcuts to dense communities, enabling global collaborative signals to propagate to cold users.

# 4 EXPERIMENTS

We validate our analysis using three real-world recommendation datasets: MovieLens-20M, Netflix, and Million Song Dataset and the A/B testing on an Amazon streaming platform. Specifically:

- First, we assess benefits of the proposed personalized item alignment approach compared to vanilla VAE-based CF (Multi-VAE (Liang et al., 2018)) on these benchmark datasets.
- Second, we provide visualizations of the learned latent space under three conditions: VAE without masking, VAE with masking, and VAE with PIA, to support our theoretical analysis.
- Finally, we conducted ablation studies across user groups segmented by interaction count to validate the effectiveness of global collaborative signals.

## 4.1 EFFECTIVENESS OF PERSONALIZED ITEM ALIGNMENT

**Public dataset.** Table 1 presents the performance of our framework, which adds personalized item alignment to Multi-VAE, and RecVAE (Shenbin et al., 2020) on the MovieLens-20M, Netflix and Million Song datasets respectively. We follow the preprocessing procedure from (Liang et al., 2018). The detailed data preprocessing steps and train/validation/test split methodology are presented in Section C.1. Our code for reproducibility is publicly available at `https://github.com/amazon-science/PIAVAE`.

Table 1: Our method (with PIA) achieves the best performance for MovieLens and Netflix Prize datasets while having the 3rd rank for Million Song. The best results are highlighted in bold.

| Model | MovieLens-20M | | | Netflix Prize | | | Million Song | | |
|---|---|---|---|---|---|---|---|---|---|
| | Recall @20 | Recall @50 | NDCG @100 | Recall @20 | Recall @50 | NDCG @100 | Recall @20 | Recall @50 | NDCG @100 |
| **Matrix factorization & Linear regression** | | | | | | | | | |
| Popularity | 0.162 | 0.235 | 0.191 | 0.116 | 0.175 | 0.159 | 0.043 | 0.068 | 0.058 |
| EASE | 0.391 | 0.521 | 0.420 | 0.362 | 0.445 | 0.393 | **0.333** | **0.428** | **0.389** |
| MF | 0.367 | 0.498 | 0.399 | 0.335 | 0.422 | 0.369 | 0.258 | 0.353 | 0.314 |
| WMF | 0.362 | 0.495 | 0.389 | 0.321 | 0.402 | 0.349 | 0.211 | 0.312 | 0.257 |
| GRALS | 0.376 | 0.505 | 0.401 | 0.335 | 0.416 | 0.365 | 0.201 | 0.275 | 0.245 |
| PLRec | 0.394 | 0.527 | 0.426 | 0.357 | 0.441 | 0.390 | 0.286 | 0.383 | 0.344 |
| WARP | 0.310 | 0.448 | 0.348 | 0.273 | 0.360 | 0.312 | 0.162 | 0.253 | 0.210 |
| LambdaNet | 0.395 | 0.534 | 0.427 | 0.352 | 0.441 | 0.386 | 0.259 | 0.355 | 0.308 |
| **Nonlinear autoencoders**: MLP for encoder | | | | | | | | | |
| CDAE | 0.391 | 0.523 | 0.418 | 0.343 | 0.428 | 0.376 | 0.188 | 0.283 | 0.237 |
| RaCT | 0.403 | 0.543 | 0.434 | 0.357 | 0.450 | 0.392 | 0.268 | 0.364 | 0.319 |
| Multi-VAE | 0.395 | 0.537 | 0.426 | 0.351 | 0.444 | 0.386 | 0.266 | 0.364 | 0.316 |
| Multi-VAE + PIA | 0.408 | 0.546 | 0.437 | 0.360 | 0.448 | 0.392 | 0.275 | 0.372 | 0.326 |
| **Uplift (%)** | 3.29 | 1.68 | 2.58 | 2.56 | 0.90 | 1.55 | 3.38 | 2.20 | 3.16 |
| **Nonlinear autoencoders**: densely connected layers for encoder | | | | | | | | | |
| RecVAE | 0.414 | 0.553 | 0.442 | 0.361 | 0.452 | 0.394 | 0.276 | 0.374 | 0.326 |
| RecVAE + PIA | **0.417** | **0.556** | **0.446** | **0.365** | **0.454** | **0.396** | 0.278 | 0.376 | 0.329 |
| **Uplift (%)** | 0.72 | 0.54 | 0.90 | 1.01 | 0.44 | 0.51 | 0.72 | 0.54 | 0.92 |

The results demonstrate that PIA consistently improves the performance over the base VAE recommenders, in terms of nDCG and Recall. It also exhibits competitive performance across the three datasets considered, with RecVAE+PIA being the top performing approach on MovieLens-20M and Netflix datasets, and the 3rd performing approach on Million Song dataset.

**A/B testing on an Amazon streaming platform.** On the basis of offline results, we run one week of A/B testing in September 2025 for the Multi-VAE + PIA algorithm on one streaming platform of Amazon. The approach was implemented as an offline system, with weekly training considering a 3-month window for collecting streaming behavior, and daily inference for active customers. The personalized scores computed daily include about 25 millions of users and 4000 movies. The performance of our system was evaluated on 2 movie cards present on the Homepage and on the Movie page. 50% of the customers were exposed to the algorithm (treatment group) for the duration

of the experiment, while the rest was exposed to the baseline algorithm (control group). As shown in Table 2, our approach outperformed the control group significantly, with improved performance on the card click rate by $117\% - 267\%$ (per daily view) and $123\% - 283\%$ (per daily user view). For statistical validation, we report the p-value which evaluates the mean difference between control and treatment groups, as well as the Bayesian probability that the mean difference is positive, considering a Normal-Normal conjugate historic prior which allows for closed form solutions to the posterior distribution. For click rate metrics we observed p-value $= 0.000$ and $(\text{prob} > 0) = 1.000$; and for playtime, we observed p-value $= 0.000$ and $(\text{prob} > 0) = 0.997$. Since its launch, we have observed the performance of the ML system to be remarkably stable in playtime and click rate metrics.

## 4.2 LATENT SPACE VISUALIZATION

We present t-SNE visualizations (Maaten & Hinton, 2008) of the latent spaces learned under three settings: ① clean input, ② input masking, and ③ input masking with personalized item alignment, to examine the correspondence between the geometry of the input space and the latent space.

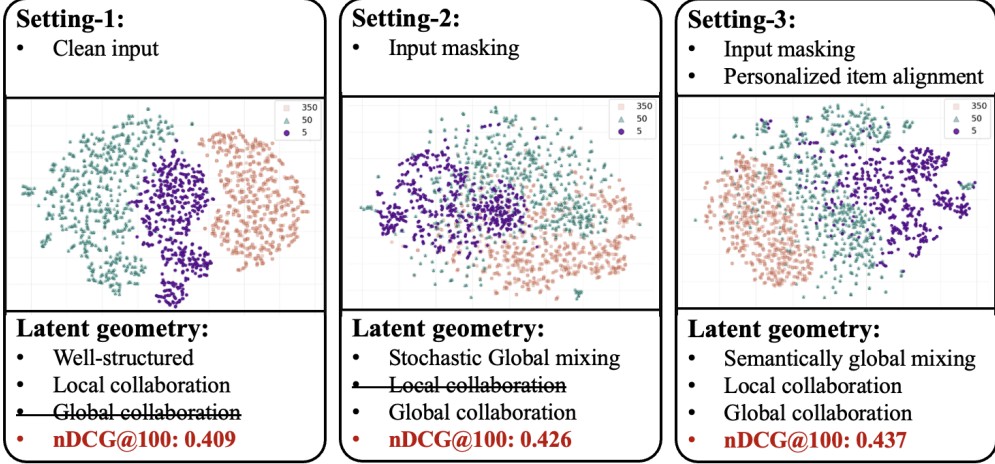

Figure 2: t-SNE visualization of the latent representations for three user groups differentiated by the number of interactions from ML-20M dataset. Purple, Teal, and Orange denote users with 5, 50, and 350 interactions, respectively, correspond to different VAE model configurations. We select group of users with 5, 50, and 350 interactions as they are clearly separated (i.e., L1) on the input space.

**Setup:** We focus on user cohorts with 5, 50, and 350 interactions to clearly contrast local versus global collaboration. Note that standard set-based distances (Hamming) inflate cross-cohort dissimilarity: even two 5-interaction users with disjoint histories are closer to each other than any 5–50 pair, even when the 50-interaction user subsumes the 5-interaction user's items; the same pattern holds for the 50–350 cohorts.

**Setting-1**: As illustrated in Figure 2, when masking is disabled and interaction counts differ substantially, the learned representations are cleanly segregated by cohort. This indicates that the model in this setting primarily leverages local collaboration and has limited ability to capture global collaborative signals. Moreover, the latent geometry mis-aligns with the global structure of the input space: the 350-interaction cluster lies closer to the 5-interaction cluster than to the 50-interaction cluster which contrary to the expected ordering, where $\text{distance}(350, 50) < \text{distance}(350, 5)$.

**Setting-2**: the representations from different cohorts become stochastically entangled, which encourages global sharing. However, within-cohort structure is more diffuse, weakening local collaboration. Despite this trade-off, **Setting-2** substantially outperforms **Setting-1** (nDCG@100$= 0.426$ vs. $0.409$), demonstrating the benefit of encouraging global collaboration.

**Setting-3** augments masking with PIA, which ① helps the VAE remain discriminative under input corruption, yielding a more structured latent space and ② promotes globally consistent user representations. The resulting latent manifold is both well organized and globally aligned, exhibiting smooth transitions from the 5-, to 50-, to 350- interaction cohorts. This balance of local and global collaboration yields the best performance (nDCG@100 $= 0.437$).

### 4.3 PERFORMANCE OF USER GROUPS WITH DIFFERENT NUMBER OF INTERACTIONS

Table 2: Offline and online results on an Amazon streaming platform.

| | Model | Recall | | nDCG |
|---|---|---|---|---|
| | | @20 | @50 | @100 |
| **Offline** | Multi-VAE | 0.592 | 0.288 | 0.386 |
| | Multi-VAE+PIA | 0.609 | 0.302 | 0.405 |
| | **Uplift (%)** | 2.87 | 4.88 | 5.13 |

| | Model | Playtime (sec) per user view | Click Rate (%) per view | Click Rate (%) per user view |
|---|---|---|---|---|
| | **Home Card** | | | |
| **Online** | Control Group | 27.7 | 4.4 | 5.3 |
| | Multi-VAE+PIA | 74.6 | 9.5 | 12.0 |
| | **Uplift (%)** | 169 | 117 | 123 |
| | **Movie Card** | | | |
| | Control Group | 16.8 | 3.4 | 4.2 |
| | Multi-VAE+PIA | 102.6 | 12.5 | 16.2 |
| | **Uplift (%)** | 509 | 267 | 283 |

Table 3: Results across user groups for MovieLens20M.

| Group | Model | Recall | | nDCG |
|---|---|---|---|---|
| | | @20 | @50 | @100 |
| [5–10] | Multi-VAE | 0.461 | 0.625 | 0.317 |
| | Multi-VAE + PIA | 0.473 | 0.629 | 0.323 |
| | **Uplift (%)** | 2.72 | 0.55 | 1.63 |
| [11–50] | Multi-VAE | 0.421 | 0.595 | 0.429 |
| | Multi-VAE + PIA | 0.424 | 0.598 | 0.434 |
| | **Uplift (%)** | 0.86 | 0.49 | 0.13 |
| [51–100] | Multi-VAE | 0.313 | 0.478 | 0.497 |
| | Multi-VAE + PIA | 0.314 | 0.479 | 0.502 |
| | **Uplift (%)** | 0.26 | 0.09 | 0.85 |
| [100+] | Multi-VAE | 0.418 | 0.386 | 0.474 |
| | Multi-VAE + PIA | 0.435 | 0.393 | 0.486 |
| | **Uplift (%)** | 4.09 | 0.72 | 2.57 |

Our proposed framework provides both a well-structured latent space and the capacity to capture global collaborative signals. As a result, we expect it to benefit users across groups, including cold-start, neutral, and warm-start users. In particular, cold-start performance Xu et al. (2022); Monteil et al. (2024); Liang et al. (2025) is of particular importance in industry settings. To assess this, we partition the test-set users based on their number of interactions and evaluate the performance of our method within each group.

As shown in Table 3, our framework improves performance for all user groups. Notably, the cold-start group (within 5 to 10 interactions) and the warm-start group (more than 100 interactions) benefit the most. This can be attributed to the inherent challenges each group faces: cold-start users have limited historical data, making recommendation difficult, while warm-start users, often found in the long tail of the user distribution, typically lack sufficient collaborative overlap. Our framework addresses both issues by enhancing access to global collaborative signals.

## 5 CONCLUSION

In this work, we analyzed how collaboration emerges in VAE–CF and showed that it is fundamentally governed by *latent* proximity: SGD updates are shared within a data-dependent *sharing radius*, clean inputs bias the model toward *local* collaboration, and global signals can be induced by either the $\beta$-KL/prior pathway (near-uniform contraction of latent distances, with collapse risk if over-used) or by input masking (stochastic neighborhood mixing with potential drift). Guided by these insights, we introduced *Personalized Item Alignment* (PIA), a training-only regularizer that attaches learnable item anchors and softly pulls masked encodings toward each user's anchor centroid. PIA preserves instance information, stabilizes the geometry under masking, and promotes *semantically grounded* global mixing without adding test-time overhead. Empirically, PIA improves over vanilla VAE–CF on standard benchmarks and in an A/B test on a large-scale streaming platform, with ablations across user-activity strata and latent-space visualizations corroborating the theory.

**Limitations.** The benefits of capturing global collaborative signals still depend heavily on how well the input masking is designed (most current research relies on Bernoulli masking). If the masking is too noisy, even with alignment mechanisms, the model may struggle to learn meaningful representations. Therefore, a promising direction for future research is to explore more effective masking strategies that better support global collaboration.

### ACKNOWLEDGMENTS

We would like to thank Vy Vo for the valuable discussions during the preparation of this paper.

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

## THE USE OF LARGE LANGUAGE MODELS

We used a large language model (ChatGPT) to help with editing this paper. It was only used for simple tasks such as fixing typos, rephrasing sentences for clarity, and improving word choice. All ideas, experiments, and analyses were done by the authors, and the use of LLMs does not affect the reproducibility of our work.

We also used ChatGPT to assist with proof verification and theorem refinement. Our workflow involved providing initial drafts to ChatGPT, which would then suggest improvements to the mathematical presentation and formatting. We subsequently edited and refined these suggestions.

## APPENDIX

This supplementary material provides a summary of common notations, detailed experimental settings, and proofs for the theoretical results stated in the main paper. It is organized as follows:

- We summarize common notation in Section A.
- We present the Related Work in Section B.
- Detailed experimental settings and implementation details are described in Section C.
- The pseudo-code of the algorithm is provided in Section C.2.
- Additional experiments on parameter sensitivity analysis are presented in Section D.
- We present all proofs relevant to the theory developed in our paper in Section E.

## A  NOTATION SUMMARY

Table 4: Table of Notations

| Symbol | Description |
|---|---|
| **Users and Items Input Data** | |
| $U, I$ | Number of users and number of items |
| $\mathbf{x}_u = [\mathbf{x}_{u1}, \mathbf{x}_{u2}, \ldots, \mathbf{x}_{uI}]$ | $I$-dimensional binary vector (the $u$-th row of $\mathbf{X}$); $\mathbf{x}_{ui} = 1$ implies that user $u$ has a positive interaction with item $i$; $\mathbf{x}_{ui} = 0$ indicates otherwise |
| $\mathbf{b} \in \{0,1\}^I$ | a binary mask, i.e., $\mathbf{b} \sim \mathrm{Bern}(\rho)^I$ |
| $\mathbf{x}_h = \mathbf{x} \odot \mathbf{b}$ | user's partial interaction history |
| $S_{\mathbf{x}} = \{\forall i \leq I : \mathbf{x}_i = 1\}$ | a set of positive items from user $\mathbf{x}$ |
| **VAE Models** | |
| $\phi, \theta$ | VAE encoder $p_\theta$ and decoder parameters $q_\phi$ |
| $q_\phi, p_\theta$ | $\phi$-parameterized and $\theta$-parameterized neural networks |
| $\mathbf{z}$ | Latent space of the VAE, e.g., $(\boldsymbol{\mu}, \boldsymbol{\sigma}^2) = q_\phi(\mathbf{x}_h)$, $\boldsymbol{\epsilon} \sim \mathcal{N}(\mathbf{0}, \mathbf{I})$, $\mathbf{z} = \boldsymbol{\mu} + \boldsymbol{\epsilon}\boldsymbol{\sigma}$ |
| $\mathbf{e}_i \in \mathbb{R}^d$ | learnable item embedding in latent space (same dimension as $\mathbf{z}$) and $E = \{\mathbf{e}_i \in \mathbb{R}^d\}_{i=1}^I$ |
| **Theoretical Constants and Bounds** | |
| $N_\delta(u) = \{v : \|\mathbf{x}_u - \mathbf{x}_v\|_1 \leq \delta\}$ | input-space neighborhood |
| $\|\cdot\|_1$ | $L_1$ norm (sum of absolute values) |
| $W_1(.,.)$ | 1-Wasserstein distance |

## B  RELATED WORKS

Motivated by our theoretical analysis, both $\beta$-weighted KL regularization and input masking can promote *global* collaboration, albeit with different trade-offs. Prior work has largely focused on

controlling the KL term, primarily along two lines: (i) scheduling the $\beta$ factor in the regularizer and (ii) adopting more flexible priors.

$\beta$-**scheduling.** Liang et al. (Liang et al., 2018) introduce a $\beta$-scaling factor to modulate the strength of the regularization $\mathrm{KL}(q_\phi(\mathbf{z} \mid \mathbf{x}) \, \| \, p(\mathbf{z}))$, while Long et al. (Long et al., 2019) propose gradually increasing this weight over training to mitigate posterior collapse.

**Flexible priors and architectures.** Several works replace the standard normal prior with richer alternatives to better match the data. Examples include VampPrior and its hierarchical variants (e.g., HVamp) (Tomczak & Welling, 2018; Kim & Suh, 2019), as well as implicit or learned priors (Walker et al., 2022). RecVAE (Shenbin et al., 2020) combines a redesigned encoder–decoder, a composite prior, input-dependent $\beta(\mathbf{x})$ rescaling, alternating training, and a non-denoising decoder. Other lines incorporate user-dependent priors (Karamanolakis et al., 2018) or impose an arbitrary target prior via adversarial training (Zhang et al., 2018).

To our knowledge, ours is the first work to systematically analyze the collaboration mechanisms in VAE-based CF, showing that both $\beta$-weighted KL regularization and input masking can promote *global* collaboration. **In contrast to prior works**, guided by our theoretical analysis, we propose a regularization scheme that addresses the issues induced by input masking, mitigating the loss of local collaboration while preserving its benefits for global alignment.

## C  EXPERIMENTAL SETTINGS

### C.1  DATASET

We validate our analysis using three real-world recommendation datasets: MovieLens-20M[2], Netflix[3] and Million Song (MSD) [4], where each record consists of a user-item pair along with a rating that the user has given to the item. We follow the preprocessing procedure from MultVAE (Liang et al., 2018). For MovieLens-20M and Netflix, we retain users who have rated at least five movies and treat ratings of four or higher as positive interactions. For MSD, we keep only users with at least 20 songs in their listening history and songs that have been listened to by at least 200 users.

Table 5: Dataset statistics.

|  | ML-20M | Netflix | MSD |
| --- | --- | --- | --- |
| # of users | 136,677 | 463,435 | 571,355 |
| # of items | 20,108 | 17,769 | 41,140 |
| # of interactions | 10.0M | 56.9M | 33.6M |
| % of interactions | 0.36% | 0.69% | 0.14% |
| # of held-out users | 10,000 | 40,000 | 50,000 |

The user data is split into training, validation, and test sets as presented in Table 5. For every user in the training set, we utilize all interaction history, whereas for users in the validation or test set, a fraction of the history (80%) is used to predict the remaining interaction.

### C.2  IMPLEMENTATION DETAILS

**Hyperparameters.** We use a batch size of 500 and train the model for 200 epochs using the Adam optimizer with a learning rate of $1 \times 10^{-3}$ across all experiments. The specific hyperparameters $\lambda_\mathrm{A}$, $\rho$, and $\lambda_\mathrm{scale}$ are selected based on validation performance. An ablation study on the sensitivity of $\rho$, $\lambda_\mathrm{scale}$ and $\lambda_\mathrm{A}$ with respect to model performance is provided in Section D.1.

**Algorithm.** The model is trained by optimizing the objective defined in Eq. (8). However, for the hyperparameter $\lambda_\mathrm{A}$, which controls the strength of the personalized item alignment regularization,

---

[2]https://grouplens.org/datasets/movielens/20m/

[3]https://www.kaggle.com/netflix-inc/netflix-prize-data

[4]http://millionsongdataset.com

instead of consider it as fixed hyper-parameter, we gradually increase its value during training .Specifically, we use the validation set to monitor whether the latent space is getting trapped in a local optimum i.e., when the validation performance does not improve after $\rho$ consecutive epochs. If such a case is detected, we increase $\lambda_{\text{A}}$ by a scaling factor $\lambda_{\text{scale}}$, as detailed in Algorithm 1.

---

**Algorithm 1** Personalized Item Alignment VAE

---

1: **Initialize**:
   Models: encoder $q_\phi$, decoder $p_\theta$ and item-embeddings $E$.
   Hyper-parameters: $\lambda_{\text{A}}$, $\lambda_{\text{scale}}$ and $\rho$.
   Variables: *best_val_epoch* := 0, *best_val_ndcg* := 0
2: **for** *epoch* **in** *n_epochs* **do**
3:    // Training
4:    **for** *iter* **in** *iterations* **do**
5:       Sample a mini-batch $\mathbf{x}$
6:       $\mathbf{z} \sim q_\phi(\cdot \mid \mathbf{x})$ // using reparametrization trick
7:       Update $q_\phi, p_\theta$ and $E$ based on $\mathcal{L}_{\text{PIA-VAE}}(\mathbf{x}; \theta, \phi)$ in Equation (8)
8:    **end for**
9:    //Validation
10:    Compute nDCG@K on validation set: *epoch_ndcg*
11:    **if** *epoch_ndcg* > *best_val_ndcg* **then**
12:       *best_val_ndcg* := *epoch_ndcg*
13:       *best_val_epoch* := *epoch*
14:    **end if**
15:    // Increase $\lambda_{\text{A}}$ if training is stuck in a local optimum, i.e., when the validation performance does not improve after $\rho$ consecutive epochs.
16:    **if** *best_val_epoch* < *epoch* + $\rho$ **then**
17:       $\lambda_{\text{A}} := \lambda_{\text{scale}} \times \lambda_{\text{A}}$
18:    **end if**
19: **end for**
20: **Return:** the optimal encoder $q_\phi$ and decoder $p_\theta$ at *best_val_epoch*.

---

### C.3 BASELINES

We have selected following models as baselines:

- *Matrix factorization* (MF); we consider MF trained with ALS with uniform weights (Hu et al., 2008), which is a simple and computationally efficient baseline, and also weighted matrix factorization (wMF) (Hu et al., 2008);

- *Regularization based on item-item interactions*; here we selected GRALS (Rao et al., 2015) that employs graph regularization;

- *Linear models*; we have chosen full-rank models EASE Steck (2019) and a low-rank model PLRec (Sedhain et al., 2016);

- *Nonlinear autoencoders*; here we consider the shallow autoencoder CDAE (Wu et al., 2016), variational autoencoder MultVAE (Liang et al., 2018), and its successors: RaCT (Lobel et al., 2019) and RecVAE (Shenbin et al., 2020).

## D ADDITIONAL EXPERIMENTS

### D.1 PARAMETER SENSITIVITY ANALYSIS ON $\lambda_{\text{A}}$ $\rho$ AND $\lambda_{\text{SCALE}}$

Note that the hyperparameter $\lambda_{\text{A}}$ was introduced in our main objective, where it controls the strength of the personalized item alignment regularization. In contrast, $\rho$ and $\lambda_{\text{scale}}$ are two hyperparameters introduced in our algorithm to dynamically adjust $\lambda_{\text{A}}$ during training. Specifically, we use $\rho$ and $\lambda_{\text{scale}}$ to gradually increase the value of $\lambda_{\text{A}}$. We monitor the performance on the validation set to determine whether the latent space becomes trapped in a local optimum i.e., when the performance

does not improve after $\rho$ epochs. If such a situation is detected, we increase $\lambda_A$ by multiplying it with the scaling factor $\lambda_{\text{scale}}$.

In this section, we analyze the sensitivity of $\lambda_A$, $\rho$, and $\lambda_{\text{scale}}$ with respect to model performance. We fix the value of $\lambda_A = \{2, 4, 8\}$ and vary the values of $\rho$ and $\lambda_{\text{scale}}$. Note that $\rho$ is used to detect local optima by checking whether the validation performance fails to improve for $\rho$ consecutive epochs. We evaluate $\rho$ in the range from 3 to 15. The hyperparameter $\lambda_{\text{scale}}$ controls the rate at which $\lambda_A$ increases. Since a large scaling factor may destabilize training, we test $\lambda_{\text{scale}}$ values in the range from 1.0 to 3.0. 1.0 mean there will no scaling.

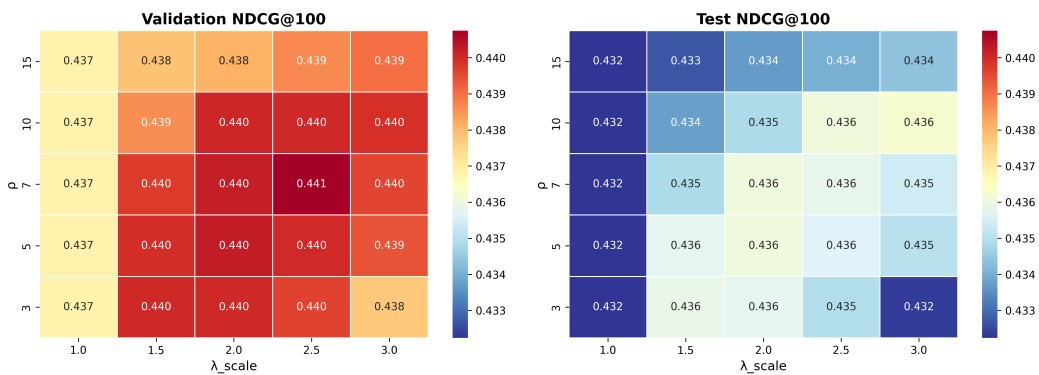

Figure 3: Parameter sensitivity analysis on $\rho$ and $\lambda_{\text{scale}}$ of PIA-VAE on MovieLens-20M. Fix $\lambda_A = 2.0$ different values of $\rho \in \{3, 5, 7, 10, 15\}$ and $\lambda_{\text{scale}} \in \{1.0, 1.5, 2, 0, 2.5, 3.0\}$

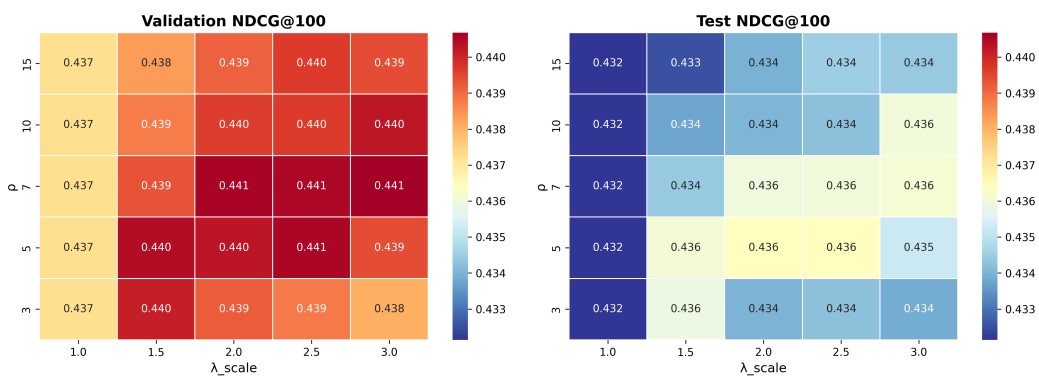

Figure 4: Parameter sensitivity analysis on $\rho$ and $\lambda_{\text{scale}}$ of PIA-VAE on MovieLens-20M. Fix $\lambda_A = 4.0$ different values of $\rho \in \{3, 5, 7, 10, 15\}$ and $\lambda_{\text{scale}} \in \{1.0, 1.5, 2, 0, 2.5, 3.0\}$

Figures 3, 4, and 5 show the performance measured by nDCG@100 on the MovieLens-20M dataset. It can be observed that the validation nDCG@100 remains relatively stable across $\lambda_A$ values, for $\lambda_{\text{scale}}$ values in the range 1.5 to 2.3 and for $\rho$ values in the range 5 to 10. Based on this analysis, we set $\rho = 5$ and $\lambda_{\text{scale}} = 2$ for all experimental settings.

**Analysis on $\lambda_A$**   To futher analyze the sensitivity of $\lambda_A$, we fix the values of $\rho = 5$ and $\lambda_{\text{scale}} = 2$ (based on previous analyses) and vary the value of $\lambda_A$.

As described in the experimental setup, the hyperparameter $\lambda_A$ is selected using nDCG@100 on the validation set as the evaluation metric across all experiments. Figure 6 presents the validation nDCG@100 for different values of $\lambda_A$, along with the corresponding test nDCG@100, Recall@20 and Recall@50 on MovieLens-20M. It can be seen that the validation and test performances are aligned, and the results also indicate that PIA-VAE is generally robust to variations when $\lambda_A > 0$.

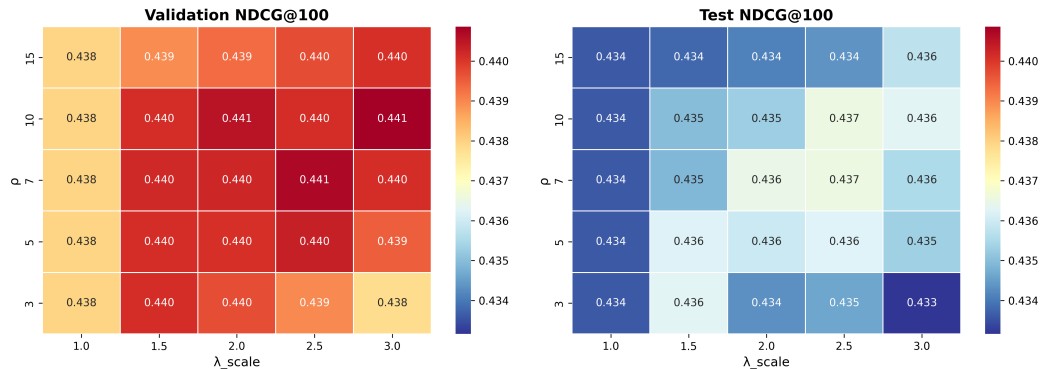

Figure 5: Parameter sensitivity analysis on $\rho$ and $\lambda_{\text{scale}}$ of PIA-VAE on MovieLens-20M. Fix $\lambda_A = 8.0$ different values of $\rho \in \{3, 5, 7, 10, 15\}$ and $\lambda_{\text{scale}} \in \{1.0, 1.5, 2, 0, 2.5, 3.0\}$

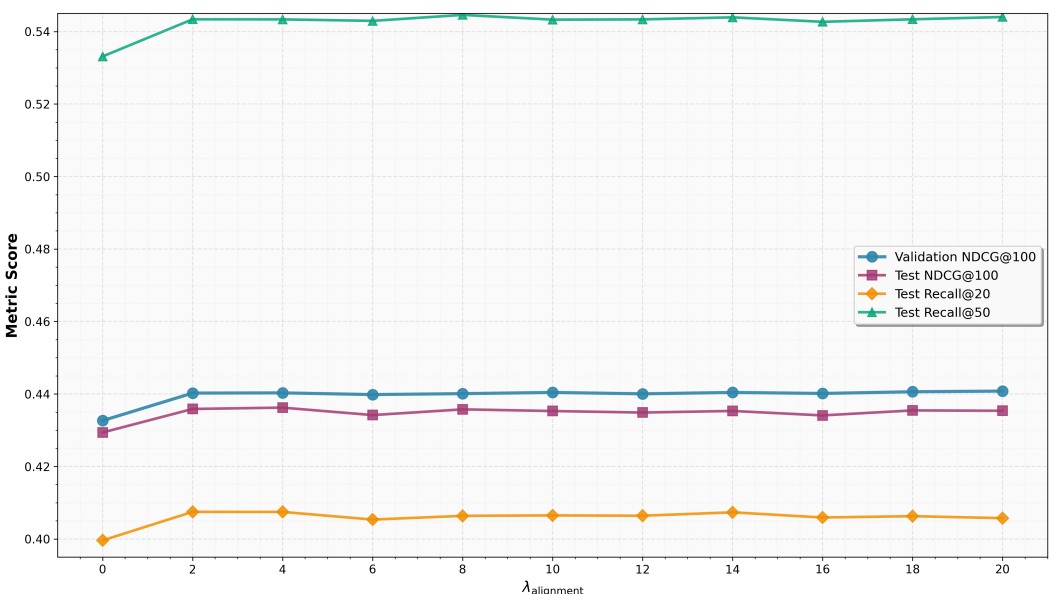

Figure 6: Performances on MovieLens-20M dataset with different $\lambda_A$

### D.2 COMPUTATIONAL INFRASTRUCTURE AND RUNNING TIME

**Infrastructure** All experiments, including our approach and the baseline models, were conducted on an `ml.g5.2xlarge` AWS EC2 instance equipped with an NVIDIA A10G GPU.

Table 6: Average training time per epoch (in seconds).

|  | VAE | VAE + Alignment |
|---|---|---|
| MovieLens-20M | 8.207 | 9.516 (15.95%) |
| Netflix | 22.193 | 25.722 (15.90%) |

**Running Time** Table 6 reports the average training time per epoch (in seconds) for the base VAE model and our proposed PIA-VAE. The addition of the alignment component in PIA-VAE results in a training overhead of approximately 15.9% compared to the base VAE.

# E  THEORETICAL DEVELOPMENT

In this Section, we present all proofs relevant to theory developed in our paper.

## E.1  PROOF OF THEOREM 2.3

**Theorem E.1** (Latent-$W_1$ Sharing Radius)**.** *Assume decoder gradient is Lipschitz in z such that for any given* $\mathbf{z} \sim q_\phi(\cdot \mid \mathbf{x}_u)$, *uniformly in* $\mathbf{x}$ *and for all* $\mathbf{z}'$, $\|\nabla_\theta \ell_\theta(\mathbf{x}, \mathbf{z}) - \nabla_\theta \ell_\theta(\mathbf{x}, \mathbf{z}')\| \le L_{\theta z} \|\mathbf{z} - \mathbf{z}'\|$. *Let*

$$\mathcal{L}_u(\theta) := \mathbb{E}_{q_\phi(\cdot \mid \mathbf{x}_u)}[\ell_\theta(\mathbf{x}_u, \mathbf{z})], \qquad g_u(\theta) := \nabla_\theta \mathcal{L}_u(\theta),$$

*and define the content-mismatch term*

$$\Delta_x(u, v) := \left\| \mathbb{E}_{q_\phi(\cdot \mid \mathbf{x}_v)} \big[ \nabla_\theta \ell_\theta(\mathbf{x}_u, \mathbf{z}) - \nabla_\theta \ell_\theta(\mathbf{x}_v, \mathbf{z}) \big] \right\|.$$

*For one SGD step on user* $v$ *with step size* $\eta > 0$, *setting* $\theta^+ = \theta - \eta\, g_v(\theta)$, *we have:*

$$\mathcal{L}_u(\theta^+) - \mathcal{L}_u(\theta) \le -\eta \|g_u(\theta)\|^2 + \eta \|g_u(\theta)\| \Big( L_{\theta z} W_1\big(q_\phi(\cdot \mid \mathbf{x}_u), q_\phi(\cdot \mid \mathbf{x}_v)\big) + \Delta_x(u, v) \Big) + O(\eta^2).$$

*In particular, the step on user* $v$ *strictly decreases* $\mathcal{L}_u$ *to first order whenever*

$$W_1\big(q_\phi(\cdot \mid \mathbf{x}_u), q_\phi(\cdot \mid \mathbf{x}_v)\big) < r_{\mathrm{share}}(u, v; \theta) := \frac{\|g_u(\theta)\| - \Delta_x(u, v)}{L_{\theta z}}.$$

*Proof.* For shorthand, write $q_u := q_\phi(\cdot \mid \mathbf{x}_u)$ and $q_v := q_\phi(\cdot \mid \mathbf{x}_v)$.

A first-order Taylor expansion gives

$$\mathcal{L}_u(\theta - \eta g_v) = \mathcal{L}_u(\theta) - \eta \langle g_u(\theta), g_v(\theta) \rangle + O(\eta^2). \tag{11}$$

Then, we decompose the inner product:

$$-\langle g_u, g_v \rangle = -\|g_u\|^2 - \langle g_u, g_v - g_u \rangle \le -\|g_u\|^2 + \|g_u\| \|g_v - g_u\|.$$

Next, we bound

$$\|g_v - g_u\| = \left\| \mathbb{E}_{q_v} \nabla_\theta \ell_\theta(\mathbf{x}_v, \mathbf{z}) - \mathbb{E}_{q_u} \nabla_\theta \ell_\theta(\mathbf{x}_u, \mathbf{z}) \right\|$$

$$\le \underbrace{\left\| \mathbb{E}_{q_v} \big[ \nabla_\theta \ell_\theta(\mathbf{x}_u, \mathbf{z}) - \nabla_\theta \ell_\theta(\mathbf{x}_v, \mathbf{z}) \big] \right\|}_{= \Delta_x(u,v)} + \underbrace{\left\| \mathbb{E}_{q_v} \nabla_\theta \ell_\theta(\mathbf{x}_u, \mathbf{z}) - \mathbb{E}_{q_u} \nabla_\theta \ell_\theta(\mathbf{x}_u, \mathbf{z}) \right\|}_{\le L_{\theta z} W_1(q_v, q_u)}.$$

For the last inequality, let $\psi(\mathbf{z}) := \nabla_\theta \ell_\theta(\mathbf{x}_u, \mathbf{z})$. By assumption decoder gradient is Lipschitz, $\psi$ is $L_{\theta z}$-Lipschitz in $\mathbf{z}$; for any coupling $\pi$ of $(q_u, q_v)$,

$$\left\| \mathbb{E}_{q_u}[\psi] - \mathbb{E}_{q_v}[\psi] \right\| = \left\| \int (\psi(\mathbf{z}) - \psi(\mathbf{z}'))\, d\pi(\mathbf{z}, \mathbf{z}') \right\| \le L_{\theta z} \int \|\mathbf{z} - \mathbf{z}'\|\, d\pi \le L_{\theta z} W_1(q_u, q_v).$$

Combining the bounds and substituting into Eq. (11) yields the stated result, and the strict-decrease condition follows by inspecting the coefficient of $\eta$. $\qquad\square$

## E.2  PROOF OF THEOREM 2.5

Before proving the theorem, we recall some inequalities:

- The Kantorovich-Rubinstein dual form (Santambrogio, 2015):

$$W_1(\mu, \nu) = \sup_{\substack{f: \mathcal{X} \to \mathbb{R} \\ \mathrm{Lip}(f) \le 1}} \int_{\mathcal{X}} f\, d(\mu - \nu), \tag{12}$$

- The Donsker-Varadhan variational formula (Donsker & Varadhan, 1975): for any measurable $g$ with $\int e^g\, dp < \infty$,

$$\mathrm{KL}(\nu\|p) \ge \int g\, d\nu - \log \int e^g\, dp. \tag{13}$$

- The Bobkov–Götze/Talagrand $T_1(C)$ inequality (Bobkov & Götze, 1999) is known to be equivalent to the following *sub-Gaussian moment generating function (mgf) bound* for Lipschitz functions: $p$ satisfies the $T_1(C)$ inequality with constant $C > 0$, then for every 1-Lipschitz $f$ and every $\lambda \in \mathbb{R}$,

$$\log \int \exp\big(\lambda(f - \mathbb{E}_p f)\big)\, dp \;\le\; \frac{C\lambda^2}{2}. \tag{14}$$

**Lemma E.2.** *Assume the encoder is $L_\phi$-Lipschitz, i.e., $\|q_\phi(\mathbf{x}_u) - q_\phi(\mathbf{x}_v)\| \le L_\phi \|\mathbf{x}_u - \mathbf{x}_v\|$ for all $\mathbf{x}_u, \mathbf{x}_v \in \{0,1\}^I$ then $W_1\big(q_\phi(\cdot \mid \mathbf{x}_u), q_\phi(\cdot \mid \mathbf{x}_v)\big) \;\le\; L_\phi \,\|\mathbf{x}_v - \mathbf{x}_u\|_1$ for all $\mathbf{x}_u, \mathbf{x}_v \in \{0,1\}^I$.*

*Proof.* First, since $q_\phi(\mathbf{z}|\mathbf{x}_i) = \mathcal{N}\big(\mu_\phi(\mathbf{x}_i), \mathrm{diag}(\sigma_\phi^2(\mathbf{x}_i))\big)$, the Wasserstein-2 distance $W_2(q_\phi(\mathbf{z}|\mathbf{x}_1), q_\phi(\mathbf{z}|\mathbf{x}_2))$ has the following closed form:

$$W_2(q_\phi(z|\mathbf{x}_u), q_\phi(z|\mathbf{x}_v))^2 = \|\mu_\phi(\mathbf{x}_u) - \mu_\phi(\mathbf{x}_v)\|^2 + \|\sigma_\phi(\mathbf{x}_u) - \sigma_\phi(\mathbf{x}_v)\|^2, \tag{15}$$

which, combined with the definition $Q_\phi(x) = \begin{bmatrix} \mu_\phi(\mathbf{x}) \\ \sigma_\phi(\mathbf{x}) \end{bmatrix}$, yields

$$\|Q_\phi(\mathbf{x}_u) - Q_\phi(\mathbf{x}_v)\|^2 = W_2(q_\phi(\mathbf{z}|\mathbf{x}_1), q_\phi(\mathbf{z}|\mathbf{x}_2))^2. \tag{16}$$

Since $Q_\phi$ is $L_\phi$-Lipschitz continuous, we have $\|Q_\phi(\mathbf{x}_u) - Q_\phi(\mathbf{x}_v)\| \le L_\phi \|\mathbf{x}_u - \mathbf{x}_v\|$, and

$$W_2(q_\phi(\mathbf{z} \mid \mathbf{x}_u), q_\phi(\mathbf{z} \mid \mathbf{x}_v)) \le L_\phi \|\mathbf{x}_u - \mathbf{x}_v\|. \tag{17}$$

Since $W_1 \le W_2$, we have $W_1(q_\phi(\mathbf{z} \mid \mathbf{x}_u), q_\phi(\mathbf{z} \mid \mathbf{x}_v)) \le L_\phi \|\mathbf{x}_u - \mathbf{x}_v\|$. $\qquad\square$

**Theorem E.3.** *Assume $p_\theta(\mathbf{x} \mid \mathbf{z})$ is a regular exponential family with sufficient statistics $T(\mathbf{x})$, natural parameter $\eta(\mathbf{z})$ and log-partition $A$. Let $\alpha(\mathbf{z}) = \frac{q_\phi(\mathbf{z}|\mathbf{x}_1)}{q_\phi(\mathbf{z}|\mathbf{x}_1)+q_\phi(\mathbf{z}|\mathbf{x}_2)}$ on $\{q_1 + q_2 > 0\}$. Then*

$$\min_{\eta(\cdot)} \left\{ \sum_{i=1}^2 \mathbb{E}_{q_\phi(\cdot|\mathbf{x}_i)}\big[ -\log p_\theta(\mathbf{x}_i \mid \mathbf{z})\big] + \beta \sum_{i=1}^2 \mathrm{KL}\big(q_\phi(\mathbf{z} \mid \mathbf{x}_i) \,\|\, p(\mathbf{z})\big) \right\} \tag{18}$$

$$= \mathrm{C} + \int \big(q_\phi(\mathbf{z} \mid \mathbf{x}_1) + q_\phi(\mathbf{z} \mid \mathbf{x}_2)\big)\, \Delta_{A^*}\big(\mathbf{x}_1, \mathbf{x}_2; \alpha(\mathbf{z})\big)\, d\mathbf{z} + \beta \sum_{i=1}^2 \mathrm{KL}\big(q_\phi(\mathbf{z} \mid \mathbf{x}_i) \,\|\, p(\mathbf{z})\big),$$

*where $A^*$ is the convex conjugate of $A$, $C$ is independent of $\eta(\cdot)$, and*

$$\Delta_{A^*}\big(\mathbf{x}_1, \mathbf{x}_2; \alpha\big) = \alpha\, A^*\big(T(\mathbf{x}_1)\big) + (1-\alpha)\, A^*\big(T(\mathbf{x}_2)\big) - A^*\big(\alpha T(\mathbf{x}_1) + (1-\alpha)T(\mathbf{x}_2)\big) \;\ge 0,$$

*with equality iff either $(q_\phi(\cdot \mid \mathbf{x}_1), q_\phi(\cdot \mid \mathbf{x}_2) = 0$ almost everywhere or $T(\mathbf{x}_1) = T(\mathbf{x}_2)$.*

*Additionally, assume the prior $p$ satisfies the Bobkov–Götze/Talagrand $T_1(C)$ inequality (Bobkov & Götze, 1999) with constant $C > 0$ (e.g., Normal prior $p = \mathcal{N}(0, \sigma^2 I)$):*

$$W_1\big(q_\phi(\mathbf{z} \mid \mathbf{x}_1),\, q_\phi(\mathbf{z} \mid \mathbf{x}_2)\big) \;\le\; \sqrt{2C\, \mathrm{KL}\big(q_\phi(\mathbf{z} \mid \mathbf{x}_1) \,\|\, p(\mathbf{z})\big)} + \sqrt{2C\, \mathrm{KL}\big(q_\phi(\mathbf{z} \mid \mathbf{x}_2) \,\|\, p(\mathbf{z})\big)}. \tag{19}$$

*Proof of Eq. (18).* We write the conditional likelihood in its exponential-family form

$$p_\theta(\mathbf{x} \mid \mathbf{z}) = h(\mathbf{x})\, \exp\big(\langle T(\mathbf{x}), \eta(\mathbf{z})\rangle - A(\eta(\mathbf{z}))\big),$$

so that

$$-\log p_\theta(\mathbf{x}_i \mid \mathbf{z}) = A(\eta(\mathbf{z})) - \langle T(\mathbf{x}_i), \eta(\mathbf{z})\rangle - \log h(\mathbf{x}_i).$$

Let $q_i(\mathbf{z}) := q_\phi(\mathbf{z} \mid \mathbf{x}_i)$, $i = 1, 2$, and define

$$Q(\mathbf{z}) := q_1(\mathbf{z}) + q_2(\mathbf{z}), \qquad \alpha(\mathbf{z}) := \begin{cases} \dfrac{q_1(\mathbf{z})}{Q(\mathbf{z})}, & Q(\mathbf{z}) > 0, \\ \text{arbitrary in } [0,1], & Q(\mathbf{z}) = 0, \end{cases}$$

and the $\alpha$-mixture of sufficient statistics

$$\overline{T}_\alpha(\mathbf{z}) := \alpha(\mathbf{z})\, T(\mathbf{x}_1) + (1 - \alpha(\mathbf{z}))\, T(\mathbf{x}_2).$$

(Any choice of $\alpha$ on $\{Q = 0\}$ is immaterial since all integrands below are multiplied by $Q$.)

Summing the two reconstruction terms and using $\int q_i = 1$,

$$\sum_{i=1}^{2} \mathbb{E}_{q_i}\big[ - \log p_\theta(\mathbf{x}_i \mid \mathbf{z})\big] = \int \Big(\sum_{i=1}^{2} q_i(\mathbf{z})\Big) A(\eta(\mathbf{z}))\, d\mathbf{z} - \int \langle T(\mathbf{x}_1)q_1(\mathbf{z}) + T(\mathbf{x}_2)q_2(\mathbf{z}),\, \eta(\mathbf{z})\rangle d\mathbf{z} - \sum_{i=1}^{2} \log h(\mathbf{x}_i)$$

$$= \int Q(\mathbf{z})\Big(A(\eta(\mathbf{z})) - \langle \overline{T}_\alpha(\mathbf{z}), \eta(\mathbf{z})\rangle\Big)\, d\mathbf{z} - \sum_{i=1}^{2} \log h(\mathbf{x}_i).$$

Recall the convex conjugate $A^*$ and Fenchel–Young inequality (Fenchel, 1949; Rockafellar, 1970):

$$A^*(y) := \sup_\eta\{\langle y, \eta\rangle - A(\eta)\}, \qquad A(\eta) - \langle y, \eta\rangle \geq -A^*(y),$$

with equality when $y \in \partial A(\eta)$; in the regular (Legendre) case (Rockafellar, 1970), $A$ is essentially smooth and strictly convex, so $y = \nabla A(\eta)$ is the unique equality condition and $A^*$ is strictly convex on its (convex) effective domain.

Because the integrand is separable in $\mathbf{z}$ and $Q(\mathbf{z}) \geq 0$, minimizing the integral over all measurable $\eta(\cdot)$ reduces to pointwise minimization:

$$\min_{\eta(\cdot)} \int Q(\mathbf{z})\Big(A(\eta(\mathbf{z})) - \langle \overline{T}_\alpha(\mathbf{z}), \eta(\mathbf{z})\rangle\Big)\, d\mathbf{z} = \int Q(\mathbf{z}) \min_\eta \big\{A(\eta) - \langle \overline{T}_\alpha(\mathbf{z}), \eta\rangle\big\}\, d\mathbf{z}$$

$$= -\int Q(\mathbf{z})\, A^*(\overline{T}_\alpha(\mathbf{z}))\, d\mathbf{z}.$$

Hence

$$\min_{\eta(\cdot)} \sum_{i=1}^{2} \mathbb{E}_{q_i}\big[ - \log p_\theta(\mathbf{x}_i \mid \mathbf{z})\big] = -\int Q(\mathbf{z})\, A^*(\overline{T}_\alpha(\mathbf{z}))\, d\mathbf{z} - \sum_{i=1}^{2} \log h(\mathbf{x}_i). \tag{20}$$

(When $\overline{T}_\alpha(\mathbf{z})$ lies in the interior of $\mathrm{dom}(A^*)$, the minimizer is $\eta^*(\mathbf{z}) = \nabla A^*(\overline{T}_\alpha(\mathbf{z}))$; equivalently, $\nabla A(\eta^*(\mathbf{z})) = \overline{T}_\alpha(\mathbf{z})$.)

Add and subtract the quantity

$$\int Q(\mathbf{z})\Big(\alpha(\mathbf{z})\, A^*(T(\mathbf{x}_1)) + (1 - \alpha(\mathbf{z}))\, A^*(T(\mathbf{x}_2))\Big)\, d\mathbf{z} = \sum_{i=1}^{2} A^*(T(\mathbf{x}_i)) \underbrace{\int q_i(\mathbf{z})\, d\mathbf{z}}_{=1},$$

and collect the terms independent of $\eta(\cdot)$ into the constant

$$\mathrm{C} := -\sum_{i=1}^{2} \big(\log h(\mathbf{x}_i) + A^*(T(\mathbf{x}_i))\big).$$

Using Eq. (20), we obtain

$$\min_{\eta(\cdot)} \sum_{i=1}^{2} \mathbb{E}_{q_i}\big[ - \log p_\theta(\mathbf{x}_i \mid \mathbf{z})\big] = \mathrm{C} + \int Q(\mathbf{z})\Big(\alpha(\mathbf{z})\, A^*(T(\mathbf{x}_1)) + (1 - \alpha(\mathbf{z}))\, A^*(T(\mathbf{x}_2)) - A^*(\overline{T}_\alpha(\mathbf{z}))\Big)\, d\mathbf{z}$$

$$= \mathrm{C} + \int Q(\mathbf{z})\, \Delta_{A^*}\big(\mathbf{x}_1, \mathbf{x}_2; \alpha(\mathbf{z})\big)\, d\mathbf{z}, \tag{21}$$

where

$$\Delta_{A^*}\big(\mathbf{x}_1, \mathbf{x}_2; \alpha\big) := \alpha\, A^*(T(\mathbf{x}_1)) + (1 - \alpha)\, A^*(T(\mathbf{x}_2)) - A^*(\alpha T(\mathbf{x}_1) + (1 - \alpha)T(\mathbf{x}_2)).$$

By convexity of $A^*$, $\Delta_{A^*}(\mathbf{x}_1, \mathbf{x}_2; \alpha) \geq 0$ for all $\alpha \in [0, 1]$ (Jensen gap). In the regular (Legendre) case, $A^*$ is strictly convex on its effective domain, so $\Delta_{A^*}(\mathbf{x}_1, \mathbf{x}_2; \alpha) = 0$ iff either

- $T(\mathbf{x}_1) = T(\mathbf{x}_2)$, in which case the three arguments of $A^*$ coincide, or

- $\alpha \in \{0, 1\}$, i.e. $Q(\mathbf{z})\alpha(\mathbf{z})\big(1 - \alpha(\mathbf{z})\big) = 0$ for $Q$-a.e. $\mathbf{z}$. Equivalently, the posteriors have disjoint supports w.r.t. the measure $Q(\mathbf{z})\,d\mathbf{z}$ (on each point with $Q > 0$ exactly one of $q_1, q_2$ is zero).

(If $A^*$ were affine on the segment $[T(\mathbf{x}_1), T(\mathbf{x}_2)]$, equality could also occur with $0 < \alpha < 1$, but strict convexity rules this out unless $T(\mathbf{x}_1) = T(\mathbf{x}_2)$.)

Finally, the regularizer $\beta \sum_{i=1}^{2} \mathrm{KL}\big(q_\phi(\mathbf{z} \mid \mathbf{x}_i) \,\|\, p(\mathbf{z})\big)$ does not depend on $\eta(\cdot)$, hence it carries through unchanged. Combining with Eq. (21) yields Eq. (18). $\qquad\square$

*Proof of Eq. (19).* By the triangle inequality for $W_1$,

$$W_1\big(q_\phi(\cdot \mid \mathbf{x}_1), q_\phi(\cdot \mid \mathbf{x}_2)\big) \;\leq\; W_1\big(q_\phi(\cdot \mid \mathbf{x}_1), p\big) \;+\; W_1\big(q_\phi(\cdot \mid \mathbf{x}_2), p\big). \tag{22}$$

Thus it suffices to show that for any probability $\nu$ with $\mathrm{KL}(\nu\|p) < \infty$,

$$W_1(\nu, p) \;\leq\; \sqrt{2C\,\mathrm{KL}(\nu\|p)}. \tag{23}$$

Fix a 1-Lipschitz $f$ and $\lambda > 0$. Apply Eq. (13) with $g = \lambda\big(f - \mathbb{E}_p f\big)$ to obtain

$$\mathrm{KL}(\nu\|p) \;\geq\; \lambda \int \big(f - \mathbb{E}_p f\big)\,d\nu \;-\; \log \int \exp\big(\lambda(f - \mathbb{E}_p f)\big)\,dp.$$

Using the mgf bound Eq. (14) gives

$$\int f\,d(\nu - p) \;\leq\; \frac{1}{\lambda}\,\mathrm{KL}(\nu\|p) \;+\; \frac{C\lambda}{2}.$$

Optimizing the right-hand side over $\lambda > 0$ yields the minimizer $\lambda^\star = \sqrt{2\,\mathrm{KL}(\nu\|p)/C}$, and the minimum value

$$\frac{1}{\lambda^\star}\,\mathrm{KL}(\nu\|p) \;+\; \frac{C\lambda^\star}{2} \;=\; \sqrt{2C\,\mathrm{KL}(\nu\|p)}.$$

Therefore, for every 1-Lipschitz $f$,

$$\int f\,d(\nu - p) \;\leq\; \sqrt{2C\,\mathrm{KL}(\nu\|p)}.$$

Taking the supremum over all 1-Lipschitz $f$ and invoking Eq. (12) gives exactly Eq. (23).

Applying Eq. (23) to $\nu = q_\phi(\cdot \mid \mathbf{x}_1)$ and to $\nu = q_\phi(\cdot \mid \mathbf{x}_2)$ and combining with the triangle inequality in Eq. (22) yields Eq. (19). $\qquad\square$

### E.3 Proof of Theorem 2.6

**Theorem E.4** (Masked input: contraction and expansion)**.** *Let $\mathbf{x}_1, \mathbf{x}_2 \in \{0, 1\}^I$ be binary inputs. Let $b_{\mathbf{x}_1}, b_{\mathbf{x}_2} \sim \mathrm{Bern}(\rho)^I$ be independent masks and set $\mathbf{x}_1' = \mathbf{x}_1 \odot b_{\mathbf{x}_1}$ and $\mathbf{x}_2' = \mathbf{x}_2 \odot b_{\mathbf{x}_2}$. Write $h = \|\mathbf{x}_1 - \mathbf{x}_2\|_1$ and $s = \langle \mathbf{x}_1, \mathbf{x}_2 \rangle$ (so $h$ is the number of disagreeing coordinates and $s$ the count of shared 1's). For any $\delta > 0$, define $T_\delta = \lceil \delta \rceil - 1$ and $U_\delta = \lceil \delta \rceil$. Let $D' := \|\mathbf{x}_1' - \mathbf{x}_2'\|_1$. Then:*

***Contraction.***

$$\Pr\big[D' < \delta\big] \;\geq\; \big(\rho^2 + (1 - \rho)^2\big)^s \sum_{k=0}^{\min\{h, T_\delta\}} \binom{h}{k} \rho^k (1 - \rho)^{h-k}. \tag{24}$$

***Expansion.***

$$\Pr\big[D' \geq \delta\big] \geq \sum_{k=U_\delta}^{s} \binom{s}{k} \big(2\rho(1 - \rho)\big)^k \big(1 - 2\rho(1 - \rho)\big)^{s-k}. \tag{25}$$

*Proof.* Partition coordinates into

$$H := \{j : x_{1j} \neq x_{2j}\}, \quad |H| = h, \qquad S := \{j : x_{1j} = x_{2j} = 1\}, \quad |S| = s.$$

For $j \in H$, exactly one of $(x_{1j}, x_{2j})$ equals 1. After masking, the post-mask difference at $j$ equals 1 iff the unique 1 is kept, with probability $\rho$. Hence

$$Y := \sum_{j \in H} \mathbf{1}\{\text{post-mask difference at } j = 1\} \sim \text{Binomial}(h, \rho).$$

For $j \in S$, both entries are 1. The post-mask difference equals $|b_{1j} - b_{2j}|$, which is 1 iff the masks disagree; this happens with probability $\Pr(b_{1j} \neq b_{2j}) = \rho(1 - \rho) + (1 - \rho)\rho = 2\rho(1 - \rho)$. Thus

$$Z := \sum_{j \in S} |b_{1j} - b_{2j}| \sim \text{Binomial}\big(s, 2\rho(1 - \rho)\big).$$

Independence of masks across coordinates implies $Y \perp Z$, and the masked distance decomposes as

$$D' = \|\mathbf{x}_1' - \mathbf{x}_2'\|_1 = Y + Z.$$

**Contraction.** Because $D'$ is integer-valued, $D' < \delta$ is equivalent to $D' \leq T_\delta$. Consider the event $\mathcal{E} := \{Z = 0\} \cap \{Y \leq T_\delta\}$. On $\mathcal{E}$ we have $D' = Y + Z \leq T_\delta$, hence

$$\Pr[D' < \delta] \geq \Pr[\mathcal{E}] = \Pr[Z = 0] \Pr[Y \leq T_\delta]$$

by independence of $Y$ and $Z$. Now $\Pr[Z = 0] = (\Pr[b_{1j} = b_{2j}])^s = (\rho^2 + (1 - \rho)^2)^s$, and

$$\Pr[Y \leq T_\delta] = \sum_{k=0}^{\min\{h, T_\delta\}} \binom{h}{k} \rho^k (1 - \rho)^{h-k}.$$

Multiplying the two factors yields Eq. (24).

**Expansion.** Using $D' = Y + Z$ with $Y \perp Z$ and $U_\delta = \lceil \delta \rceil$,

$$\Pr[D' \geq \delta] = \Pr[Y + Z \geq U_\delta] = \sum_{m=0}^{h} \Pr[Y = m] \ \Pr[Z \geq U_\delta - m]$$

$$= \sum_{m=0}^{h} \binom{h}{m} \rho^m (1 - \rho)^{h-m} \sum_{k=\max\{U_\delta - m, 0\}}^{s} \binom{s}{k} \big(2\rho(1 - \rho)\big)^k \big(1 - 2\rho(1 - \rho)\big)^{s-k},$$

.

Finally, since $D' = Y + Z \geq Z$,

$$\Pr[D' \geq \delta] \geq \Pr[Z \geq U_\delta] = \sum_{k=U_\delta}^{s} \binom{s}{k} \big(2\rho(1 - \rho)\big)^k \big(1 - 2\rho(1 - \rho)\big)^{s-k},$$

which gives Eq. (25). This completes the proof. $\qquad\square$

### E.4  PROOF OF PROPOSITION 3.1

**Proposition E.5.** *Assume the encoder posterior is diagonal-Gaussian,* $q_\phi(\mathbf{z} \mid \mathbf{x}_h) = \mathcal{N}\big(\boldsymbol{\mu}_\phi(\mathbf{x}_h), \text{diag}(\boldsymbol{\sigma}_\phi^2(\mathbf{x}_h))\big)$. *Let the item centroid be* $\bar{\mathbf{e}}_\mathbf{x} := \frac{1}{|S_\mathbf{x}|} \sum_{i \in S_\mathbf{x}} \mathbf{e}_i$, *then*

$$\mathcal{L}_\text{A}(\mathbf{x}_h, \mathbf{x}; \phi, E) = \underbrace{\left\| \boldsymbol{\mu}_\phi(\mathbf{x}_h) - \bar{\mathbf{e}}_\mathbf{x} \right\|^2}_{\text{align mean to item centroid}} + \underbrace{\text{tr} \Sigma_\phi(\mathbf{x}_h)}_{\text{variance shrinkage}} + const(\mathbf{x}, E), \qquad (26)$$

*where* $\Sigma_\phi(\mathbf{x}_h) = \text{diag}(\boldsymbol{\sigma}_\phi^2(\mathbf{x}_h))$ *and* $const(\mathbf{x}, E) = \frac{1}{|S_\mathbf{x}|} \sum_{i \in S_\mathbf{x}} \|\mathbf{e}_i\|_2^2 - \|\bar{\mathbf{e}}_\mathbf{x}\|_2^2$.

*Proof.* Write $q_\phi(\mathbf{z} \mid \mathbf{x}_h) = \mathcal{N}\big(\boldsymbol{\mu}_\phi(\mathbf{x}_h), \Sigma_\phi(\mathbf{x}_h)\big)$ with $\Sigma_\phi(\mathbf{x}_h) = \mathrm{diag}(\boldsymbol{\sigma}_\phi^2(\mathbf{x}_h))$, we have:

$$\mathbb{E}\big[\|\mathbf{z} - \mathbf{e}_i\|_2^2\big] = \big\|\boldsymbol{\mu}_\phi(\mathbf{x}_h) - \mathbf{e}_i\big\|_2^2 + \mathrm{tr}\,\Sigma_\phi(\mathbf{x}_h).$$

Averaging over $i \in S_{\mathbf{x}}$ yields

$$\mathcal{L}_{\mathrm{A}}(\mathbf{x}_h, \mathbf{x}; \phi, E) = \frac{1}{|S_{\mathbf{x}}|} \sum_{i \in S_{\mathbf{x}}} \big\|\boldsymbol{\mu}_\phi(\mathbf{x}_h) - \mathbf{e}_i\big\|_2^2 + \mathrm{tr}\,\Sigma_\phi(\mathbf{x}_h).$$

Given the item centroid be $\bar{\mathbf{e}}_{\mathbf{x}} := \frac{1}{|S_{\mathbf{x}}|} \sum_{i \in S_{\mathbf{x}}} \mathbf{e}_i$, then, we obtain

$$\frac{1}{|S_{\mathbf{x}}|} \sum_{i \in S_{\mathbf{x}}} \big\|\boldsymbol{\mu}_\phi(\mathbf{x}_h) - \mathbf{e}_i\big\|_2^2 = \big\|\boldsymbol{\mu}_\phi(\mathbf{x}_h) - \bar{\mathbf{e}}_{\mathbf{x}}\big\|_2^2 + \frac{1}{|S_{\mathbf{x}}|} \sum_{i \in S_{\mathbf{x}}} \|\mathbf{e}_i\|_2^2 - \|\bar{\mathbf{e}}_{\mathbf{x}}\|_2^2.$$

Combining the last two displays gives

$$\mathcal{L}_{\mathrm{A}}(\mathbf{x}_h, \mathbf{x}; \phi, E) = \big\|\boldsymbol{\mu}_\phi(\mathbf{x}_h) - \bar{\mathbf{e}}_{\mathbf{x}}\big\|_2^2 + \mathrm{tr}\,\Sigma_\phi(\mathbf{x}_h) + \underbrace{\Big(\frac{1}{|S_{\mathbf{x}}|} \sum_{i \in S_{\mathbf{x}}} \|\mathbf{e}_i\|_2^2 - \|\bar{\mathbf{e}}_{\mathbf{x}}\|_2^2\Big)}_{\mathrm{const}(\mathbf{x}, E)},$$

which is exactly Eq. (26). $\qquad\square$

### E.5 PROOF OF PROPOSITION 3.2

**Proposition E.6.** *Fix $\mathbf{x}$ and its neighborhood in which the ELBO objective $\mathcal{L}_{VAE}(\mathbf{x}; \theta, \phi; \mathbf{x}_h)$, defined in Eq. (1), written as a function of the encoder mean $\boldsymbol{\mu}_\phi(\mathbf{x}_h) \in \mathbb{R}^d$, admits a quadratic approximation with mask-independent curvature*

$$H \succeq mI, \qquad \|H\|_2 \le L \qquad (0 < m \le L < \infty).$$

*Adding $\lambda_{\mathrm{A}}\|\boldsymbol{\mu}_\phi(\mathbf{x}_h) - \bar{\mathbf{e}}_{\mathbf{x}}\|_2^2$ to this objective yields an effective Hessian $H_{eff} = H + 2\lambda_{\mathrm{A}}I$. Let $\boldsymbol{\mu}^{(0)}(\mathbf{x}_h)$ be the unregularized minimizer over masks and $\boldsymbol{\mu}^{(A)}(\mathbf{x}_h)$ the minimizer with alignment. Then with $\tau = \left(\frac{L}{L + 2\lambda_{\mathrm{A}}}\right)$, we obtain the following inequalities*

$$\big\|\mathrm{Var}_{\mathbf{b}}\big[\boldsymbol{\mu}^{(A)}(\mathbf{x}_h)\big]\big\|_2 \le \tau^2 \big\|\mathrm{Var}_{\mathbf{b}}\big[\boldsymbol{\mu}^{(0)}(\mathbf{x}_h)\big]\big\|_2, \tag{27}$$

$$\mathrm{tr}\,\mathrm{Var}_{\mathbf{b}}\big[\boldsymbol{\mu}^{(A)}(\mathbf{x}_h)\big] \le \tau^2 \,\mathrm{tr}\,\mathrm{Var}_{\mathbf{b}}\big[\boldsymbol{\mu}^{(0)}(\mathbf{x}_h)\big], \tag{28}$$

$$\mathbb{E}_{\mathbf{b}}\big[\,\|\boldsymbol{\mu}^{(A)}(\mathbf{x}_h) - \bar{\mathbf{e}}_{\mathbf{x}}\|_2^2\,\big] \le \tau\,\mathbb{E}_{\mathbf{b}}\big[\,\|\boldsymbol{\mu}^{(0)}(\mathbf{x}_h) - \bar{\mathbf{e}}_{\mathbf{x}}\|_2^2\,\big]. \tag{29}$$

*Moreover, if $\mathrm{Var}_{\mathbf{b}}[\boldsymbol{\mu}^{(0)}]$ commutes with $H$ (e.g., they are simultaneously diagonalizable), then the Löwner-order contraction*

$$\mathrm{Var}_{\mathbf{b}}\big[\boldsymbol{\mu}^{(A)}(\mathbf{x}_h)\big] \preceq \tau^2 \,\mathrm{Var}_{\mathbf{b}}\big[\boldsymbol{\mu}^{(0)}(\mathbf{x}_h)\big]$$

*holds.*

*Proof.* Fix $\mathbf{x}$ and a mask $\mathbf{b}$, and let $F_{\mathbf{b}}(\boldsymbol{\mu})$ denote the (unregularized) mask-conditioned denoising objective as a function of the encoder mean $\boldsymbol{\mu} \in \mathbb{R}^d$ (all other quantities $\mathbf{x}$, the masked input $\mathbf{x}_h$, decoder parameters, are held fixed).

Choose a reference point $\tilde{\boldsymbol{\mu}}$ in a neighborhood where $F_{\mathbf{b}}$ admits a quadratic approximation with *mask-independent* curvature matrix $H$, and assume

$$H \succeq mI, \qquad \|H\|_2 \le L, \qquad 0 < m \le L < \infty.$$

Equivalently, we approximate the Hessian uniformly across masks by $\nabla^2 F_{\mathbf{b}}(\tilde{\boldsymbol{\mu}}) \approx H$.

**Quadratic surrogate.** Define the quadratic model of $F_{\mathbf{b}}$ around $\tilde{\boldsymbol{\mu}}$ by

$$J_{\mathbf{b}}(\boldsymbol{\mu}) := F_{\mathbf{b}}(\tilde{\boldsymbol{\mu}}) + \nabla F_{\mathbf{b}}(\tilde{\boldsymbol{\mu}})^{\top} (\boldsymbol{\mu} - \tilde{\boldsymbol{\mu}}) + \tfrac{1}{2} (\boldsymbol{\mu} - \tilde{\boldsymbol{\mu}})^{\top} H (\boldsymbol{\mu} - \tilde{\boldsymbol{\mu}}). \tag{30}$$

Let

$$\mathbf{g}_{\mathbf{b}} := \nabla F_{\mathbf{b}}(\tilde{\boldsymbol{\mu}}), \qquad \mathbf{a}_{\mathbf{b}} := \tilde{\boldsymbol{\mu}} - H^{-1}\mathbf{g}_{\mathbf{b}}.$$

Completing the square yields an equivalent form

$$J_{\mathbf{b}}(\boldsymbol{\mu}) = c_{\mathbf{b}} + \tfrac{1}{2} (\boldsymbol{\mu} - \mathbf{a}_{\mathbf{b}})^{\top} H (\boldsymbol{\mu} - \mathbf{a}_{\mathbf{b}}), \tag{31}$$

where the (mask-dependent) constant

$$c_{\mathbf{b}} := F_{\mathbf{b}}(\tilde{\boldsymbol{\mu}}) - \tfrac{1}{2} \mathbf{g}_{\mathbf{b}}^{\top} H^{-1}\mathbf{g}_{\mathbf{b}}$$

is independent of $\boldsymbol{\mu}$.

In particular, the unique minimizer of $J_{\mathbf{b}}$ is $\mathbf{a}_{\mathbf{b}}$:

$$\arg\min_{\boldsymbol{\mu}} J_{\mathbf{b}}(\boldsymbol{\mu}) = \mathbf{a}_{\mathbf{b}}.$$

If $F_{\mathbf{b}}$ is exactly quadratic with curvature $H$ in this neighborhood, then $\boldsymbol{\mu}^{(0)}(\mathbf{x}_h) = \mathbf{a}_{\mathbf{b}}$; otherwise, $\mathbf{a}_{\mathbf{b}}$ is the minimizer of the local quadratic approximation to $F_{\mathbf{b}}$.

Adding the alignment penalty $\lambda_{\mathrm{A}} \|\boldsymbol{\mu} - \bar{\mathbf{e}}_{\mathbf{x}}\|_2^2$ gives the first–order condition

$$(H + 2\lambda_{\mathrm{A}}I) \boldsymbol{\mu} = H \mathbf{a}_{\mathbf{b}} + 2\lambda_{\mathrm{A}}\bar{\mathbf{e}}_{\mathbf{x}}.$$

Define

$$M := (H + 2\lambda_{\mathrm{A}}I)^{-1}H, \qquad \tau := \frac{L}{L + 2\lambda_{\mathrm{A}}} \in (0, 1).$$

Then the aligned minimizer is the affine *shrinkage* of $\mathbf{a}_{\mathbf{b}}$ toward $\bar{\mathbf{e}}_{\mathbf{x}}$:

$$\boldsymbol{\mu}^{(\mathrm{A})} = M \mathbf{a}_{\mathbf{b}} + (I - M) \bar{\mathbf{e}}_{\mathbf{x}}. \tag{32}$$

**Spectral bounds on $M$.** Diagonalize $H = Q\Lambda Q^{\top}$ with $\Lambda = \mathrm{diag}(\lambda_i)$, $m \leq \lambda_i \leq L$. Then

$$M = Q \, \mathrm{diag}\Big(\frac{\lambda_i}{\lambda_i + 2\lambda_{\mathrm{A}}}\Big) Q^{\top}.$$

Let $\alpha_i := \lambda_i/(\lambda_i + 2\lambda_{\mathrm{A}}) \in (0, 1)$. It follows that

$$0 \preceq M \preceq I, \qquad \|M\|_2 = \max_i \alpha_i = \frac{\lambda_{\max}(H)}{\lambda_{\max}(H) + 2\lambda_{\mathrm{A}}} \leq \tau, \qquad \|M^2\|_2 = \|M\|_2^2 \leq \tau^2, \tag{33}$$

and, eigenwise, $\alpha_i^2 \leq \tau \, \alpha_i$, hence

$$M^2 \preceq \tau M \preceq \tau I. \tag{34}$$

**Variance contraction (operator norm).** Since $\bar{\mathbf{e}}_{\mathbf{x}}$ is mask–independent, Eq. (32) gives

$$\mathrm{Var}_{\mathbf{b}}\big[\boldsymbol{\mu}^{(\mathrm{A})}\big] = M \, \mathrm{Var}_{\mathbf{b}}\big[\mathbf{a}_{\mathbf{b}}\big] M. \tag{35}$$

Taking spectral norms and using submultiplicativity,

$$\big\| \mathrm{Var}_{\mathbf{b}}[\boldsymbol{\mu}^{(\mathrm{A})}] \big\|_2 \leq \|M\|_2^2 \big\| \mathrm{Var}_{\mathbf{b}}[\mathbf{a}_{\mathbf{b}}] \big\|_2 \leq \tau^2 \big\| \mathrm{Var}_{\mathbf{b}}[\mathbf{a}_{\mathbf{b}}] \big\|_2,$$

which is Eq. (27) upon noting $\mathrm{Var}_{\mathbf{b}}[\mathbf{a}_{\mathbf{b}}] = \mathrm{Var}_{\mathbf{b}}[\boldsymbol{\mu}^{(0)}]$.

**Variance contraction (trace).** From Eq. (35),

$$\mathrm{tr} \, \mathrm{Var}_{\mathbf{b}}\big[\boldsymbol{\mu}^{(\mathrm{A})}\big] = \mathrm{tr}\Big( \mathrm{Var}_{\mathbf{b}}[\mathbf{a}_{\mathbf{b}}] M^2 \Big) \leq \|M^2\|_2 \, \mathrm{tr} \, \mathrm{Var}_{\mathbf{b}}[\mathbf{a}_{\mathbf{b}}] \leq \tau^2 \, \mathrm{tr} \, \mathrm{Var}_{\mathbf{b}}[\mathbf{a}_{\mathbf{b}}],$$

where the inequality uses $M^2 \preceq \|M^2\|_2 I$ and the fact that for $A, B \succeq 0$, $\mathrm{tr}(AB) \leq \|B\|_2 \, \mathrm{tr}(A)$. This yields Eq. (28).

**Löwner-order contraction under commutation (Higham & Lin, 2013).** If $\mathrm{Var_b}[\mathbf{a_b}]$ commutes with $H$, then it commutes with $M$. In the common eigenbasis, write $\mathrm{Var_b}[\mathbf{a_b}] = Q\,\mathrm{diag}(v_i)\,Q^\top$ with $v_i \geq 0$. Then

$$M\,\mathrm{Var_b}[\mathbf{a_b}]\,M = Q\,\mathrm{diag}(\alpha_i^2 v_i)\,Q^\top \;\preceq\; Q\,\mathrm{diag}(\tau^2 v_i)\,Q^\top = \tau^2\,\mathrm{Var_b}[\mathbf{a_b}],$$

proving the Löwner-order bound.

**Mean–drift contraction.** From Eq. (32),

$$\boldsymbol{\mu}^{(\mathrm{A})} - \bar{\mathbf{e}}_{\mathbf{x}} = M(\mathbf{a_b} - \bar{\mathbf{e}}_{\mathbf{x}}),$$

hence

$$\begin{aligned}
\|\boldsymbol{\mu}^{(\mathrm{A})} - \bar{\mathbf{e}}_{\mathbf{x}}\|_2^2 &= (\mathbf{a_b} - \bar{\mathbf{e}}_{\mathbf{x}})^\top M^2 (\mathbf{a_b} - \bar{\mathbf{e}}_{\mathbf{x}}) \\
&\leq (\mathbf{a_b} - \bar{\mathbf{e}}_{\mathbf{x}})^\top (\tau I)(\mathbf{a_b} - \bar{\mathbf{e}}_{\mathbf{x}}) \qquad \text{(by Eq. (34))} \\
&= \tau\,\|\mathbf{a_b} - \bar{\mathbf{e}}_{\mathbf{x}}\|_2^2.
\end{aligned}$$

Taking $\mathbb{E}_{\mathbf{b}}$ gives

$$\mathbb{E}_{\mathbf{b}}\big[\|\boldsymbol{\mu}^{(\mathrm{A})} - \bar{\mathbf{e}}_{\mathbf{x}}\|_2^2\big] \;\leq\; \tau\,\mathbb{E}_{\mathbf{b}}\big[\|\boldsymbol{\mu}^{(0)} - \bar{\mathbf{e}}_{\mathbf{x}}\|_2^2\big],$$

which is Eq. (29).

$\square$

