# OpenReview forum: "On the Mechanisms of Collaborative Learning in VAE Recommenders"
_ICLR.cc/2026/Conference — ICLR 2026 Poster_

### Official Review · Reviewer_Mp4J · 2025-10-31

**Soundness:** 2
**Presentation:** 2
**Contribution:** 3
**Rating:** 2
**Confidence:** 3

**Summary:**

In this paper, the authors theoretically analyzed two existing mechanisms, $\beta$-KL regularization and input masking, on their effect of trade-off between local and global collaboration. To address shortcomings of these methods, the authors proposed Personalized Item Alignment (PIA), which stabilized the latent geometry without the risks of over regularization. The authors presented offline and online experiments and visualizations to validate the effectiveness of PIA.

**Strengths:**

This paper provides a deep theoretical dive into how collaboration occurs in VAE-CF models. The derivation of latent sharing radius (Theorem 2.3) presents a formal mechanism to understand the interplay between users when updated. The analysis of the trade-offs achieved by $\beta$-KL regularization and input masking is insightful for practitioners to understand and optimize these methods.
The proposed Personalized Item Alignment (PIA) is clean and intuitive. By introducing item anchors in latent space, PIA makes use of the semantics of user-item interaction records and mitigate the drawbacks of existing methods.

**Weaknesses:**

The experiment results are not very persuasive. The public datasets are relatively out-of-date so less convincing. The online results, though significant in uplift, draw comparison to a weak statistical baseline, which barely supports the validation.
Introducing learnable item anchors increases the number of parameters of the model. This is probably a huge gap as the size of item corpus is often quite large in realworld recommender systems. In this case, the experiments would be unfair.
The idea of introducing item anchor seems closely related to VQ-VAE, which is not cited nor discussed in this paper.
The provided visualization demonstrates the ability of PIA to yield more structured latent space under input corruption. However, there is no evidence showing the drift introduced by PIA encourages global collaboration between "far but related" users, which is the motivation of the idea.
I believe the readibility of the paper would be improved with more visualization and case studies.

**Questions:**

The effect of PIA seems to vary significantly with the encoder model, as provided in the offline results. Can the authors provide more results and discussion?

---

> ### Author Response · Authors · 2025-11-17
> **Response to the comments of Reviewer Mp4J (Part 1)**
>
> We sincerely would like to express our great gratitude to the reviewer for dedicating your time to reviewing our work. Below, we outline our efforts to address the valuable points you have raised.
>
> &nbsp;
>
> **Weakness-1: The public datasets are relatively out-of-date so less convincing.**
>
> *Reply.*
>
> Our primary rationale for using MovieLens-20M, Netflix, and MSD is that they remain *standard, actively used benchmarks*. This choice allows our results to be directly comparable to prior work and to well-established baselines. Additionally, these datasets align well with our production use case, making our findings more practically relevant.
>
> Recent papers (even new trending LLM-based methods) continue to rely on these benchmarks when studying recommender models and training strategies (e.g., [1,2,3,4,5]), indicating that they are still considered relevant and informative by the community.
>
> &nbsp;
>
> [1] Milogradskii, A., Lashinin, O., P, A., Ananyeva, M., \& Kolesnikov, S. (2024). Revisiting BPR: A Replicability Study of a Common Recommender System Baseline. Proceedings of the 18th ACM Conference on Recommender Systems.
>
> [2] Öncel, F., Penaloza, E., Wu, H., Gupta, S., Ravanelli, M., Charlin, L., \& Subakan, C. (2025). Audio Prototypical Network for Controllable Music Recommendation. 2025 IEEE 35th International Workshop on Machine Learning for Signal Processing (MLSP), 1-6.
>
> [3] Subbiah, Anushya et al. “Improved Estimation of Ranks for Learning Item Recommenders with Negative Sampling.” Proceedings of the 33rd ACM International Conference on Information and Knowledge Management (2024): n. pag.
>
> [4] Vančura, Vojtěch, Pavel Kordík, and Milan Straka. "beeFormer: Bridging the Gap Between Semantic and Interaction Similarity in Recommender Systems." Proceedings of the 18th ACM Conference on Recommender Systems. 2024.
>
> [5] Wang, Shijie, et al. "Knowledge graph retrieval-augmented generation for llm-based recommendation." Proceedings of the 63rd Annual Meeting of the Association for Computational Linguistics (Volume 1: Long Papers). 2025.
>
> &nbsp;
>
> **Weakness-2: The online results, though significant in uplift, draw comparison to a weak statistical baseline, which barely supports the validation.**
>
> *Reply.*
>
> The statistical baseline (control group) represents an operationally-validated baseline that our model and system (treatment group) must outperform to justify deployment. It is out of our control to choose the baseline used for A/B testing.
>
> For comparison with the stronger baseline multiVAE, refer to Table 2. We conducted offline evaluation using a production-realistic setup: models were trained weekly on a 3-month sliding window of streaming behavior data and deployed for daily inference on active customers.
>
> Since launching this model in production, we have observed the performance of the ML system to be remarkably stable in playtime and click rate metrics. The approach is powering the recommendation of a movie widget regularly achieving top-3 daily performance across all movie widgets at a platform level.
>
> &nbsp;
>
> **Weakness-3: Introducing learnable item anchors increases the number of parameters of the model. This is probably a huge gap as the size of item corpus is often quite large in realworld recommender systems. In this case, the experiments would be unfair.**
>
> *Reply.*
>
> **Efficient training with sparse interactions:** During training, we only look up and update anchor vectors for items that actually appear in the current mini-batch, i.e., items the batch users have interacted with. In typical recommendation domains (movies, music, e-commerce), each user interacts with at most a few hundred items, so the number of distinct items per batch is small. As a result, the *per-iteration* memory footprint and computation cost scale with the number of batch-distinct items and are effectively independent of the total item-set size.
>
> **No additional information:** The item anchors are just a standard item-embedding table learned from the same implicit feedback as the baselines; we do not use any side features or extra supervision. Thus, PIA does not introduce additional information beyond what the base models already consume.
>
> **No inference overhead:** Anchors are used only during training to shape the latent geometry and are not required at test time, so they do not increase the memory or latency of the deployed recommender.
>
> &nbsp;
>
> In summary, these design choices ensure fairness and scalability: the gains of PIA come from a better geometric organization of users and items, not from extra information or inflated capacity.

---

> ### Author Response · Authors · 2025-11-17
> **Response to the comments of Reviewer Mp4J (Part 2)**
>
> **Weakness-4: The idea of introducing item anchors seems closely related to VQ-VAE, which is not cited nor discussed in this paper.**
>
> *Reply.*
>
> Conceptually, our item anchors are closer to the broad family of learnable prototypes/embeddings (e.g., prototypical networks, metric-learning class anchors, k-means/mixture centroids, word embeddings) than to vector quantization as in VQ-VAE.
>
> Item anchor different from VQ-VAE in purpose and mechanism:
>
>
> **Purpose.**
> - VQ-VAE: Introduces a discrete codebook and performs hard quantization of latents, primarily for compression and learning discrete latent variables.
> - PIA: Are continuous, item-tied reference points used to stabilize the latent space geometry and facilitate collaborative signal propagation (users who share items become closer; items that share users cluster).
>
>
> **Mechanism.**
> - VQ-VAE: Replaces the latent representation with the nearest codeword at both training and inference, and optimizes a codebook/commitment objective.
> - PIA: Adds a *soft* alignment penalty towards users' liked item anchors *during training only*; the encoder/decoder remain standard VAE-CF at test time, and anchors are not used at inference.
>
> &nbsp;
>
> **Weakness-5: The provided visualization demonstrates the ability of PIA to yield more structured latent space under input corruption. However, there is no evidence showing the drift introduced by PIA encourages global collaboration between "far but related" users, which is the motivation of the idea. I believe the readibility of the paper would be improved with more visualization and case studies.**
>
> **Reply.*
>
> We appreciate this thoughtful question and would like to clarify how our visualization demonstrates global collaboration:
>
> &nbsp;
>
> In Figure-2, we deliberately select three user groups with very different interaction counts to represent ``far but related'' users:
>
> - Group 1 (5 interactions): sparse, low-activity users,
> - Group 2 (50 interactions): medium-activity users,
> - Group 3 (350 interactions): dense, high-activity users.
>
> These groups are far apart in the input space (large $\ell_1$ distance due to very different interaction patterns), but can still share underlying preferences.
>
> &nbsp;
>
> The key observations are:
>
> - **Without masking (setting-1):** the three groups remain well separated in the latent space, largely clustered by activity level. This indicates that collaboration is predominantly *local*, occurring within similar-activity groups.
>
> - **With masking: (setting-2 and setting-3)** the colored groups mix substantially in the latent space. Users with very different interaction counts (e.g., 5 vs. 350 items) are now positioned near each other when they share underlying preferences. This color mixing is precisely the ``far but related'' global collaboration we target: masking induces long-range sharing between distant user groups.
>
>
> While random masking already creates some global mixing, PIA (Setting-3) makes this global collaboration *more structured and semantically aligned*: users who share items are brought into tighter, more coherent neighborhoods, rather than relying on purely stochastic mixing.

---

> ### Author Response · Authors · 2025-11-17
> **Response to the comments of Reviewer Mp4J (Part 3)**
>
> **Question: The effect of PIA seems to vary significantly with the encoder model, as provided in the offline results. Can the authors provide more results and discussion?**
>
> *Reply.*
>
> Our goal is to introduce an *orthogonal* mechanism to prior work, which primarily improves VAE-CF via the KL term (e.g., $\beta$-schedules, expressive priors, hierarchical architectures). In contrast, PIA explicitly addresses the masking effect and acts on the geometric organization of users and items, while remaining lightweight and training-only.
>
> Note that the full RecVAE setup comprises an architecture change, a weighted-$\beta$ objective, and a composite prior. In our main experiments, we initially use only the architectural component to isolate the effect of PIA from RecVAE’s weighted-$\beta$ and composite prior.
>
>
> To directly address the Reviewer’s concern:
> - We additionally report results where PIA is combined with the *full* RecVAE setup (architecture + weighted-$\beta$ + composite prior).
>
> - We further include experiments where PIA is combined with HVamp (hierarchical VAE + VampPrior), demonstrating that PIA yields consistent gains on strong baselines.
>
> &nbsp;
>
> **Table: Performance of PIA with different architectures on MovieLens-20M**
>
> | Model | Recall @20 | Recall @50 | nDCG @100 |
> |-------|------------|------------|-----------|
> | Multi-VAE | 0.395 | 0.537 | 0.426 |
> | Multi-VAE + PIA | 0.408 | 0.546 | 0.437 |
> | **Uplift (%)** | 3.29 | 1.68 | 2.58 |
> | | | | |
> | RecVAE | 0.414 | 0.553 | 0.442 |
> | RecVAE + PIA | 0.419 | 0.558 | 0.448 |
> | **Uplift (%)** | 1.21 | 0.90 | 1.36 |
> | | | | |
> | HVamp | 0.413 | 0.551 | 0.445 |
> | HVamp + PIA | 0.419 | 0.556 | 0.449 |
> | **Uplift (%)** | 1.41 | 0.86 | 0.94 |
>
> These results show that PIA yields *consistent, positive gains* not only on early VAE-CF models but also when applied on top of strong, modern architectures such as RecVAE and HVamp.

---

### Official Review · Reviewer_2RrV · 2025-10-31

**Soundness:** 2
**Presentation:** 2
**Contribution:** 1
**Rating:** 4
**Confidence:** 4

**Summary:**

The authors theoretically analyze the VAE-based collaborative filtering. Specifically, the authors claim that VAE-based CFs are governed by the latent proximity. Both the Local collaboration and global collaboration mechanism has been studied. The authors introduce the concept of a sharing radius via latent Wasserstein distance and compare the distinct operational mechanisms of β-KL regularization versus input masking. Building on this analysis, they propose Personalized Item Alignment (PIA) to mitigate the adverse effects of masking. PIA has four main advantages, which are preserving instance information, stabilizing the geometric pathway, promoting meaningful global mixing, and requiring no test-time burden. Performance improvements are reported on three benchmark datasets and an Amazon A/B test.

**Strengths:**

- The approach is well-motivated theoretically.
- The claims have been supported through proofs which is provided in Appendix.
- A/B testing has been conducted on industry-level datasets with real users.

**Weaknesses:**

- The paper lacks readability. Some of the terms are quite new, and hasn’t been defined well in the manuscript. For instance, local collaboration/ global collaboration are not well established terms in VAE-CF literature. Yet the authors claim that this is the first work to systematically analyze the collaboration mechanisms.
- The statement on the base model is somewhat misleading. In line 105-106, the authors claim that the random binary mask is used across the VAE-based CF with citations. However, none of these works uses random masking.
- The literature review is missing many representative VAE-based models. The baseline models are also missing models with strong performances.
- PIA only exhibits modest improvements even on the conventional VAE CF models. These early models can be improved with less efforts compared to recent models. Is PIA applicable to recent VAE-based models?
- Computational complexity hasn't been discussed. No ablation study has been performed.

**Questions:**

- PIA targets the *average* of the item anchors that a user likes. However, when a user has diverse interests (e.g., 50% action movies, 50% romance movies), the *midpoint* between these two preferences may represent a meaningless (non-existent) preference. Isn't there a risk that this 'average' centroid distorts the representation of users with multi-modal interests?
- How does the each component: item alignment, variance shrinkage, and λ_A scheduling, contribute towards claimed improvements?
- Table 3 shows a 2.72% recall improvement for the [5-10] interaction group, but is this really due to global collaborative signals? By definition, cold-start users have small S_x, which means the item centroid ē_x has low reliability—how does PIA help in this case?

---

> ### Author Response · Authors · 2025-11-17
> **Response to the comments of Reviewer 2RrV (Weaknesses: Part 1)**
>
> We sincerely would like to express our great gratitude to the reviewer for dedicating your time to reviewing our work. Below, we outline our efforts to address the valuable points you have raised.
>
> &nbsp;
>
> **Weakness-1: The paper lacks readability. Some of the terms are quite new, and hasn’t been defined well in the manuscript. For instance, local collaboration/global collaboration are not well established terms in VAE-CF literature. Yet the authors claim that this is the first work to systematically analyze the collaboration mechanisms.**
>
> *Reply.*
>
> The notions of *local* and *global* collaboration are indeed new; our aim is precisely to provide a first systematic analysis of how VAE-CF models share information across users. To the best of our knowledge, existing VAE-CF work does not offer established terminology for these distinct collaboration patterns. We therefore view making these interaction modes explicit and precisely defined as an integral part of the paper’s methodological contribution.
>
> Please let us know if you have any suggestions to further improve the readability and presentation. We are happy to incorporate them in the final version.
>
> &nbsp;
>
> **Weakness-2: The statement on the base model is somewhat misleading. In line 105-106, the authors claim that the random binary mask is used across the VAE-based CF with citations. However, none of these works uses random masking.**
>
> *Reply.*
>
> We thank the Reviewer for this comment and agree that our wording (``random binary mask is used across VAE-based CF'') can be misleading. Our intention was to emphasize that *stochastic input corruption* (via dropout / denoising) is standard practice in VAE-CF, even if it is not always described with the same terminology as in our paper.
>
> Concretely, prior VAE-CF works apply random input perturbations during training:
>
> - Liang et al. (2018): *"Section 2.3 A taxonomy of autoencoders"* discusses denoising autoencoders and applies input dropout as a stochastic corruption mechanism.
> - Lobel et al. (2019, RaCT): *"Section 2: Background: VAEs for Collaborative Filtering"* describes the use of input dropout.
> - Shenbin et al. (2020, RecVAE): *"Section 3.2: Model Architecture"* describes perturbed inputs in the training pipeline.
> - Vanvcura  & Kordik (2021, VASP): *"Section D: Data augmentation to prevent learning identity"* uses random perturbations of the input.
> - Kim & Suh (2019, HVAMP): employ dropout in the released implementation, even if it is not heavily emphasized in the main text.
>
>
>
> In the revised version, we (i) soften the statement to say that *dropout-style random masking is commonly used* in VAE-based CF, and (ii) explicitly refer to these works as using stochastic input corruption, rather than claiming that they all employ our exact ``random binary mask'' formulation.
>
> &nbsp;
>
> References:
>
> Liang, et al. ``Variational Autoencoders for Collaborative Filtering.'' WWW 2018.
>
> Lobel, et al. ``Towards Amortized Ranking-Critical Training for Collaborative Filtering.'' ICLR 2020.
>
> Kim \& Suh. ``Enhancing VAEs for Collaborative Filtering: Flexible Priors \& Gating Mechanisms.'' RecSys 2019.
>
> Vanvcura, V., \& Kordik, P. ``Deep Variational Autoencoder with Shallow Parallel Path for Top-N Recommendation (VASP).'' ICANN 2021.
>
> Shenbin, et al. ``RecVAE: A New Variational Autoencoder for Top-N Recommendations with Implicit Feedback.'' WSDM 2020.
>
> &nbsp;
>
> **Weakness-3: The literature review is missing many representative VAE-based models. The baseline models are also missing models with strong performances.**
>
> *Reply.*
>
> We appreciate this concern and will expand the related work section to more thoroughly cover recent VAE-based CF models. In terms of experimental baselines, our study already includes **RecVAE** (Shenbin et al., 2020), which is one of the strongest reported VAE-CF models.
>
> In addition, we now include **HVamp** (Kim \& Suh, 2019), a hierarchical VAE with a VampPrior, as another strong and representative baseline (see next question for results).
>
> To the best of our knowledge, **RecVAE** and **HVamp** are amongst the most competitive VAE-CF methods, and PIA improves them by simply adding a small training-time alignment term, without modifying their architectures or test-time pipelines.

---

> ### Author Response · Authors · 2025-11-17
> **Response to the comments of Reviewer 2RrV (Weaknesses: Part 2)**
>
> **Weakness-4: PIA only exhibits modest improvements even on the conventional VAE CF models. These early models can be improved with less efforts compared to recent models. Is PIA applicable to recent VAE-based models?**
>
> *Reply.*
>
> Our goal is to introduce an *orthogonal* mechanism to prior work, which primarily improves VAE-CF via the KL term (e.g., $\beta$-schedules, expressive priors, hierarchical architectures). In contrast, PIA explicitly addresses the *masking* effect and acts on the *geometric organization* of users and items, while remaining lightweight and training-only.
>
> Note that the full RecVAE setup comprises an architecture change, a weighted-$\beta$ objective, and a composite prior. In our main experiments, we initially use only the architectural component to isolate the effect of PIA from RecVAE’s weighted-$\beta$ and composite prior.
>
> To directly address the reviewer’s concern:
> - We additionally report results where PIA is combined with the *full* RecVAE setup (architecture + weighted-$\beta$ + composite prior).
>
> - We further include experiments where PIA is combined with HVamp (hierarchical VAE + VampPrior), demonstrating that PIA yields consistent gains on strong baselines.
>
> &nbsp;
>
> **Table: Performance of PIA with different architectures on MovieLens-20M**
>
> | Model | Recall @20 | Recall @50 | nDCG @100 |
> |-------|------------|------------|-----------|
> | Multi-VAE | 0.395 | 0.537 | 0.426 |
> | Multi-VAE + PIA | 0.408 | 0.546 | 0.437 |
> | **Uplift (%)** | 3.29 | 1.68 | 2.58 |
> | | | | |
> | RecVAE | 0.414 | 0.553 | 0.442 |
> | RecVAE + PIA | 0.419 | 0.558 | 0.448 |
> | **Uplift (%)** | 1.21 | 0.90 | 1.36 |
> | | | | |
> | HVamp | 0.413 | 0.551 | 0.445 |
> | HVamp + PIA | 0.419 | 0.556 | 0.449 |
> | **Uplift (%)** | 1.41 | 0.86 | 0.94 |
>
> These results show that PIA yields *consistent, positive gains* not only on conventional VAE-CF model but also when applied on top of strong, modern architectures such as RecVAE and HVamp.
>
> Importantly, PIA is added as a small training-time alignment term and does not require modifying the underlying architectures or test-time pipelines, making it easy to integrate into existing VAE-CF systems. We are not aware of any other enhancements that provide comparable gains with such minimal integration effort.
>
> &nbsp;
>
> **Weakness-5: Computational complexity hasn't been discussed. No ablation study has been performed.**
>
> *Reply.*
>
> In the revised version, "Section E.1: Computational Infrastructure and Running Time", we additionally report the average training time per epoch (in seconds) for the base VAE model and our proposed PIA-VAE. The addition of the alignment component in PIA-VAE results in a training overhead of approximately 15.9\% compared to the base VAE.

---

> ### Author Response · Authors · 2025-11-17
> **Response to the comments of Reviewer 2RrV (Questions: Part 1)**
>
> **Question-1: PIA targets the average of the item anchors that a user likes. However, when a user has diverse interests (e.g., 50\% action movies, 50\% romance movies), the midpoint between these two preferences may represent a meaningless (non-existent) preference. Isn't there a risk that this 'average' centroid distorts the representation of users with multi-modal interests?**
>
> *Reply.*
>
> At a high level, PIA creates a *mutual benefit* between user and item representations:
> - users are pulled toward the anchors of the items they like, and
> - items that share many of the same users are pulled toward each other.
>
> This mutual organization sharpens the item geometry (items form semantic neighborhoods) and, in turn, places users at more meaningful positions relative to those neighborhoods.
>
> Geometrically, item anchors live on a semantic map where co-liked items form neighborhoods. A user’s centroid lies inside the region spanned by their liked items, not at an arbitrary point. For a multi-modal user (e.g., action + romance), the centroid sits on the shortest path between the corresponding neighborhoods, keeping the user simultaneously close to both communities.
>
> This is beneficial for collaboration: even if overall tastes are split, shared (or nearby) items still pull such users into overlapping regions, so global signals can flow from both sides.
>
> In practice, users who share more items (or items from the same neighborhoods) become closer under PIA and thus exchange more collaborative signal, rather than having their preferences collapsed into a single ``fake'' mode.
>
> &nbsp;
>
> **Question-2: Table 3 shows a 2.72\% recall improvement for the [5-10] interaction group, but is this really due to global collaborative signals? By definition, cold-start users have small $S_x$, which means the item centroid $\bar{e}_x$ has low reliability—how does PIA help in this case?**
>
> *Reply.*
>
> Here as well, the mutual organization of users and items under PIA is key.
>
> - Warm users with many interactions help structure the semantic map: items they co-like are pulled into coherent neighborhoods.
> - Cold users, even with a few clicks, are then placed into this *already-organized* map by aligning them to the anchors of their clicked items.
>
> Concretely, each clicked item is a landmark surrounded by a crowd of warm users who also liked it (and related items). Aligning the cold user toward those landmarks means that even a single shared item is often enough to snap them into the corresponding warm cluster, where many experienced users reside.
>
> Moreover, because items that share users are pulled closer together under PIA, interacting with a very similar item (same sub-genre, director, actor, franchise) also places the cold user inside the *right* neighborhood, not just the broad genre. In raw click space, such a cold user appears far from any particular warm user; in latent geometry, one or two well-placed items act as short-cuts into dense communities, enabling genuinely global collaborative signals to propagate from those warm neighborhoods to the cold user.

---

> ### Author Response · Authors · 2025-11-17
> **Response to the comments of Reviewer 2RrV (Questions: Part 2)**
>
> **Question-3: How does each component: item alignment, variance shrinkage, and $\lambda_A$ scheduling, contribute towards claimed improvements?**
>
> *Reply.*
>
> Conceptually, item alignment and variance shrinkage are two core *objective* components; and $\lambda_A$ scheduling is *algorithmic* component; we discuss them separately.
>
>
> **(1) Item alignment and variance shrinkage (objective-level).**
>
> The closed-form expansion in Proposition 1 shows that the PIA loss decomposes into:
> - (i) an *item-alignment* term that pulls the encoder mean toward the centroid of a user’s liked items, and
> - (ii) an induced *variance-shrinkage* term that slightly contracts the posterior around this aligned mean.
>
> These geometric effects are visible in Figure 2:
>
> - *Item alignment.* Comparing Setting-2 (masking only) and Setting-3 (masking + PIA), the overlap between the three user cohorts becomes more structured under PIA. In Setting-2 we observe stochastic mixing of colors, whereas in Setting-3 users from different cohorts that share preferences form clearer, semantically coherent neighborhoods rather than being randomly entangled.
> - *Variance shrinkage.* In Setting-2 the latent structure within each cohort is more diffuse, which weakens local collaboration. In contrast, in Setting-3 the within-cohort clusters are more compact, consistent with the mild variance-shrinkage effect predicted by the theory.
>
> &nbsp;
>
> **2) $\lambda_A$ scheduling (algorithmic-level).**
>
> For quantitative evidence, we provide a dedicated sensitivity analysis in Appendix E: Additional Experiments (in the revised paper).
>
> - First, we systematically vary all three hyperparameters on MovieLens-20M: we sweep $\lambda_{\text A}\in\\{2,4,8\\}$,  $\rho\in\\{3,5,7,10,15\\}$ and $\lambda_{\text{scale}}\in\\{1.0,1.5,2.0,2.5,3.0\\}$ (Fig. 3,4 and 5).
> - Based on the first analysis, we then fix $\rho=5$ and $\lambda_{\text{scale}}=2$ and sweep $\lambda_{\text A}\in\{0,2,4,\dots,20\}$, reporting validation nDCG@100 together with test nDCG@100, Recall@20, and Recall@50 (Fig. 6). Validation and test metrics are well aligned, and PIA-VAE consistently improves over the base model for a broad range of $\lambda_{\text A}>0$.
>
> &nbsp;
>
> In summary:
> - **Without $\lambda_A$ scheduling ($\lambda_{\text{scale}}=1$, so $\rho$ has no effect)**, PIA still improves performance, but the gain is small (nDCG@100 $\approx 0.432$ vs. $0.426$ for VAE).
> - **With $\lambda_A$ scheduling ($\lambda_{\text{scale}}>1$)**, the improvement becomes more pronounced. Across $\lambda_{\text{scale}}>1$ and $\rho\in \\{5,7, 10\\}$, nDCG@100 remains strong and stable in the range $\approx 0.435$ to $0.437$.

---

### Official Review · Reviewer_PHGG · 2025-11-01

**Soundness:** 3
**Presentation:** 3
**Contribution:** 3
**Rating:** 8
**Confidence:** 2

**Summary:**

This paper investigates how (local/global) collaboration emerges in VAE-CF and shows it is governed by latent proximity. They compare the mechanisms and trade-offs of β-regularization and input masking. To address the latent drift, they propose PIA, a training-only regularizer that pulls each user’s masked posterior toward the centroid of their interacted item anchors. The analysis shows that PIA aligns user posteriors with their semantic item centroids, reduces variance, and stabilizes the encoder by improving local conditioning and mitigating mask-induced drift.

**Strengths:**

1. Engaging and well-structured paper.

2. Provides clear theoretical insight into how (local/global) collaboration in VAE–CF emerges.

3. Proposes a noble regularizer (PIA) that stabilizes masking-induced noise through semantic alignment.

4. Offers rigorous analysis showing that PIA reduces latent variance and improves encoder conditioning.

The paper is theoretically solid, and the large-scale experiments convincingly support its claims.

**Weaknesses:**

I did not find any major flaws in the theoretical analysis or experimental results. The paper appears technically sound and well-executed overall. I have only a few minor questions.

**Questions:**

1. I have a question regarding the definition and effect of the item centroid $\bar e_x$ in PIA.
As I understand it, the posteriors $q_{\phi}( z |x_h^{(1)}), q_{\phi} (z|x_h^{(2)}), \dots$ from multiple masked views are softly aligned toward the same semantic centroid $\bar e_x$, making the user representation consistent across different masks.
However, for users who have interacted with a very large and diverse set of items, the centroid $\bar e _x = \frac{1}{|S_x|} \sum_{ i \in S_x} e_i$ could become an almost uniform average over many anchors and lose semantic specificity. In such cases, does PIA still provide meaningful alignment?

2. For large-scale item sets (e.g., millions of items), how does the additional anchor parameterization affect memory and training efficiency?

3. I would be interested to hear the authors’ perspective on whether using a weighted (e.g. (reflecting interaction frequency) centroid could improve alignment.

---

> ### Author Response · Authors · 2025-11-17
> **Response to the comments of Reviewer PHGG**
>
> We sincerely would like to express our great gratitude to the reviewer for dedicating your time to reviewing our work. We also thank you for your positive evaluation of our research. Below, we outline our efforts to address the valuable points you have raised.
>
> &nbsp;
>
> **Question: I have a question regarding the definition of $\bar{e}_x$ and effect of the item centroid in PIA. As I understand it, the posteriors from multiple masked views are softly aligned toward the same semantic centroid , making the user representation consistent across different masks. However, for users who have interacted with a very large and diverse set of items, the centroid $\bar e x = \frac{1}{|S_x|} \sum{ i \in S_x} e_i$ could become an almost uniform average over many anchors and lose semantic specificity. In such cases, does PIA still provide meaningful alignment?**
>
> *Reply.*
>
> At a high level, PIA induces a *mutual organization* of user and item representations:
> - users are pulled toward the anchors of the items they like, and
> - items that share many of the same users are pulled toward each other.
>
> This two-way pressure sharpens the geometry of item anchors into coherent semantic neighborhoods, and places users at locations that reflect how their personal item sets intersect these neighborhoods.
>
> From a geometric perspective, item anchors live on a semantic map where co-liked items form neighborhoods. A user’s centroid $\bar e_x$ lies inside the region spanned by their liked items, not at an arbitrary point. Even for users with large and diverse histories, the centroid remains anchored within the convex hull of their items on this map. This is still meaningful for collaboration: two users who share more items (or items from the same neighborhoods) will have centroids that are *closer*, and thus their representations become neighbors, resulting in a stronger collaborative signal based on the latent sharing radius theorem.
>
> &nbsp;
>
> **Question: For large-scale item sets (e.g., millions of items), how does the additional anchor parameterization affect memory and training efficiency?**
>
> *Reply.*
>
> **Efficient training with sparse interactions:** During training, we only look up and update anchor vectors for items that actually appear in the current mini-batch, i.e., items the batch users have interacted with. In typical recommendation domains (movies, music, e-commerce), each user interacts with at most a few hundred items, so the number of distinct items per batch is small. As a result, the *per-iteration* memory footprint and computation cost scale with the number of batch-distinct items, and are effectively independent of the total item-set size.
>
> **Memory footprint**: The item anchors form a standard $|I|\times d$ embedding table, which is the same order of magnitude as the item-embedding matrices already used in modern CF models. In practice, this table can be stored in external/parameter-server style storage and is relatively small compared to the overall model weights, even for millions of items.
>
> &nbsp;
>
> **Question: I would be interested to hear the authors’ perspective on whether using a weighted (e.g. (reflecting interaction frequency) centroid could improve alignment.**
>
> *Reply.*
>
>
> PIA pulls a user’s latent representation toward the “center’’ of the item anchors they like. For users with many or diverse interactions, a plain (unweighted) average can indeed be somewhat coarse. In principle, using a *weighted* centroid (e.g., weighting each liked item by how often or how strongly the user interacted with it) could make the alignment more faithful to the user’s true preferences by emphasizing the most semantically representative items and down-weighting weaker or generic signals.
>
> In practice, such weights could naturally be instantiated from explicit feedback (e.g., ratings in MovieLens or Netfix) or other strength indicators. However, for fair comparison with prior work and to isolate the effect of PIA itself, in our experiments we follow the standard protocol and convert ratings to binary implicit feedback, using an unweighted centroid.
>
> Additionally, in our current setup both user and item representations are learned purely from implicit feedback; incorporating item-side metadata (e.g., genres, directors, textual descriptions) to define more informed weights would be a natural and promising extension.
>
> We therefore view weighted centroids and metadata-informed weighting as interesting directions for future work, particularly in settings where interaction strength and side information are readily available and standardized evaluation protocols permit such extensions.

---

### Official Review · Reviewer_4p7K · 2025-11-01

**Soundness:** 4
**Presentation:** 4
**Contribution:** 3
**Rating:** 8
**Confidence:** 3

**Summary:**

This paper provides a theoretical and empirical analysis of collaboration mechanisms in VAE-based collaborative filtering. The authors establish that collaboration is governed by latent proximity, formalizing this concept through a derived latent sharing radius. This radius specifies the condition under which an SGD update for one user will also reduce the loss for another user, with the influence decaying as the Wasserstein distance between their latent distributions increases. The work further contrasts two pathways for enabling global collaboration: KL regularization, which acts on the objective to promote posterior overlap at the risk of representational collapse, and input masking, which operates on the data to stochastically alter input geometries, enabling global mixing but potentially introducing neighborhood drift. Guided by this analysis, the authors propose Personalized Item Alignment (PIA), a training-time regularizer that stabilizes the latent geometry under input masking by aligning a user's masked posterior toward a centroid defined by learnable item embeddings. This method aims to facilitate global collaboration in a semantically grounded way without incurring inference-time costs. Empirical validation on public benchmarks shows consistent improvements, and the method demonstrated significant metric lifts in a large-scale A/B test on an Amazon streaming platform. This paper is distinguished by its exemplary clarity in exposition, seamlessly integrating sophisticated theoretical derivations with practical algorithmic design and empirical validation.

**Strengths:**

Novel Theoretical Framework: It establishes a rigorous, interpretable theory of collaboration in VAE-CF based on a latent sharing radius, providing a geometric condition for update transfer between users.

Elegant Algorithmic Design: The proposed PIA method is a direct, low-overhead application of this theory, using learnable item anchors and a training-only regularizer without inference costs.

Extensive Empirical Support: The methodology is validated through consistent improvements on public benchmarks, detailed ablations, and supporting latent-space visualizations.

Demonstrated Practical Impact: Significant performance gains reported from a large-scale A/B test on a production platform underscore its real-world applicability.

Exemplary Exposition: The paper is exceptionally clear, seamlessly bridging complex theory, proofs, and experimental result

**Weaknesses:**

A/B test reporting is incomplete. The online results are compelling but the paper omits key statistical details (sample sizes, confidence intervals or p-values) required to assess robustness and practical significance.

Limited sensitivity analysis for hyperparameters. The geometric effects central to the paper depend on PIA hyperparameters. A more systematic ablation or robustness sweep would increase confidence that the method is stable across realistic settings.

**Questions:**

see Weaknesses

---

> ### Author Response · Authors · 2025-11-17
> **Response to the comments of Reviewer 4p7K**
>
> We sincerely would like to express our great gratitude to the reviewer for dedicating your time to reviewing our work. We also thank you for your positive evaluation of our research. Below, we outline our efforts to address the valuable points you have raised.
>
> &nbsp;
>
> **Question: A/B test reporting is incomplete. The online results are compelling but the paper omits key statistical details (sample sizes, confidence intervals or p-values) required to assess robustness and practical significance.**
>
> *Reply.*
>
> The personalized scores computed daily include about 25 millions of users and 4000 movies. 50\% of the customers were exposed to the algorithm (treatment group) for the duration of the experiment, while the rest was exposed to the baseline algorithm (control group). For our metrics (click rates and playtime), we report the p-value which evaluates the mean difference between control and treatment groups, as well as the Bayesian probability that the mean difference is positive, considering a Normal-Normal conjugate historic prior which allows for closed form solutions to the posterior distribution. For click rate metrics we observed $\text{p-value}=0.000$ and $(\text{prob}>0)=1.000$; and for playtime, we observed $\text{p-value}=0.000$ and $(\text{prob}>0)=0.997$.
>
> &nbsp;
>
> **Question: Limited sensitivity analysis for hyperparameters. The geometric effects central to the paper depend on PIA hyperparameters. A more systematic ablation or robustness sweep would increase confidence that the method is stable across realistic settings.**
>
> *Reply.*
>
> In the revised version, we have added a dedicated sensitivity analysis in Appendix E: Additional Experiments.
>
> Recall that $\lambda_{\text A}$ is the main regularization weight in the objective, while $\rho$ and $\lambda_{\text{scale}}$ are *algorithmic* hyperparameters used only to schedule $\lambda_{\text A}$ during training: when validation nDCG@100 does not improve for $\rho$ epochs, we increase $\lambda_{\text A}$ by multiplying it with $\lambda_{\text{scale}}$.
>
> In Appendix E, we systematically vary all three hyperparameters on MovieLens-20M:
> - (i) We sweep $\lambda_{\text A}\in\\{2,4,8\\}$,  $\rho\in\\{3,5,7,10,15\\}$ and $\lambda_{\text{scale}}\in\\{1.0,1.5,2.0,2.5,3.0\\}$ . The results (Fig. 3,4 and 5) show that validation nDCG@100 remains stable for $\lambda_{\text{scale}}$ in roughly $[1.5,2.3]$ and $\rho$ in $[5,10]$, and we therefore set $\rho=5$ and $\lambda_{\text{scale}}=2$ for all experiments.
>
> - (ii) Based on the first analysis, we then fix $\rho=5$ and $\lambda_{\text{scale}}=2$ and vary $\lambda_{\text A}\in  [0, 2, 4,..,20]$, reporting validation nDCG@100 together with test nDCG@100, Recall@20, and Recall@50 (Fig. 6). Validation and test metrics are well aligned, and PIA-VAE consistently improves over the base model for a broad range of $\lambda_{\text A}$.
>
> Overall, these experiments indicate that PIA's gains are robust across a range of $\lambda_{\text A}$, $\rho$, and $\lambda_{\text{scale}}>1.0$.

---

### Author Response · Authors · 2025-12-03
**Summary of Reviewers' concerns and Rebuttal**

### **Reviewer 4p7K**

---

**Q1: A/B test reporting is incomplete.**

**A:** We provide additional A/B testing settings and key statistical details.

---

**Q2: Limited sensitivity analysis for hyperparameters.**

**A:** We provide a dedicated sensitivity analysis for hyperparameters in Appendix E: Additional Experiments.

---

&nbsp;

### **Reviewer PHGG**

---

**Q1: Effect of PIA for users who have interacted with a very large and diverse set of items.**

**A:** We clarify the geometric effect of PIA and explain how it facilitates collaboration for users who have interacted with a very large and diverse set of items.

---

**Q2: How does the additional anchor parameterization affect memory and training efficiency?**

**A:** We clarify:
- No additional side information is required
- Anchors are training-only with no inference overhead
- Training is efficient with sparse interactions i.e., only active items per mini-batch are loaded

---

&nbsp;

### **Reviewer 2RrV**

---

**Q1: Local collaboration/global collaboration are not well-established terms in VAE-CF literature. Yet the authors claim this is the first work to systematically analyze collaboration mechanisms.**

**A:** We are the first to systematically analyze collaboration mechanisms in VAE-CF. Existing literature lacks established terminology for these distinct patterns; therefore, explicitly defining these interaction modes is an integral part of our methodological contribution.

---

**Q2: The literature review is missing many representative VAE-based models. The baseline models are also missing models with strong performance.**

**A:** RecVAE and HVamp are among the most competitive VAE-CF methods. Our study includes RecVAE, and we now provide additional results on HVamp.

---

**Q3: Computational complexity hasn't been discussed.**

**A:** We now report the average training time per epoch (in seconds) in Section E.1.

---

**Q4: How does each component work: item alignment, variance shrinkage, scheduling?**

**A:** We provide a dedicated sensitivity analysis for each component in Appendix E: Additional Experiments.

---

**Q5: Effect of PIA for users with diverse interests and cold-start users.**

**A:** We clarify the geometric effect of PIA and explain how it facilitates collaboration for users with diverse interests and cold-start users.

---

&nbsp;

### **Reviewer Mp4J**

---

**Q1: The public datasets are relatively out-of-date.**

**A:** We clarify that Netflix, MovieLens-20M, and the Million Song Dataset are well-established benchmarks that remain widely used in contemporary work, and we cite several recent papers (2024–2025) demonstrating their continued adoption.

---

**Q2: The effect of PIA seems to vary significantly with the encoder model.**

**A:** We clarified experimental settings and added experiments with two strong baselines:
- PIA + full-setting RecVAE (densely connected encoder),
- PIA + HVamp (hierarchical VAE).

Both baselines demonstrate consistent performance improvements.

---

**Q3: Additional parameters due to learnable item anchors make the comparison unfair.**

**A:** We clarify:
- no additional side information is required,
- anchors are training-only with no inference overhead,
- training is efficient with sparse interactions i.e., only active items per mini-batch are loaded.

---

**Q4: There is no evidence showing the drift introduced by PIA encourages global collaboration between "far but related" users, which is the motivation of the idea.**

**A:** We respectfully clarify:
- PIA does not introduce drift, it alleviates the drift caused by masking.
- Our analysis shows global collaboration arises from masking or strong $\beta$-KL regularization, as clearly demonstrated in our visualization.

---

---

### Author Response · Authors · 2025-12-03
**Contribution Summary**

We sincerely thank the Reviewers for their thoughtful reviews and the Area Chair for taking on the extra responsibility and workload resulting from the recent review reset. Below, we provide a concise summary of our paper's contributions, reviewers' concerns, and our responses.

&nbsp;

### **Main Contribution**

We provide a theoretical analysis of how collaboration occurs in VAE-CF models.

&nbsp;

**Section 2.2:** We show that collaboration emerges in VAE-CF is governed by latent proximity (Theorem 2.3).
To further understand collaboration mechanisms, we examine the correspondence between input space geometry and latent space geometry. We consider two settings:
- Local collaborative signal: users near in input space
- Global collaborative signal: users far in input space but related (i.e., interact with the same items)

&nbsp;

**Section 2.3:** We investigate how local/global collaboration emerges and show that:
- The original VAE favors local collaboration
- KL regularization and input masking can encourage global collaboration, but each has drawbacks:
  - KL regularization weakens the reconstruction signal
  - Input masking causes latent drift, making update-gradients shared among nearby users noisy and inconsistent

&nbsp;

**Section 2.4:** We link this geometric behavior to actionable insights via information theory.
- We show that previous methods improve VAE-CF by addressing KL regularization issues (e.g., $\beta$-scheduling, expressive priors).
- We demonstrate a complementary view: improving VAE-CF by addressing input masking problems, leading to our proposed PIA.

&nbsp;

**Section 3:** We propose **Personalized Item Alignment (PIA)**. By introducing item anchors in latent space, PIA leverages the semantics of user-item interaction records to stabilize geometry and promote meaningful global mixing. We provide theoretical results showing that PIA addresses latent drift and the noisy, inconsistent update-gradients shared among nearby users.

&nbsp;

**General comment:** Three reviewers (4p7K, PHGG and Mp4j) agree that the paper is engaging and well-structured, providing clear theoretical insights into how local and global collaboration emerge in VAE-CF. The approach is reported to be theoretically well-motivated, clean, and intuitive.

---

### Meta-Review · Area_Chair_zyug · 2026-01-06

**Summary:**

The reviewers consistently express positive attitudes to this submission, for example: (1) elegant algorithmic design; (2) The paper is theoretically solid and well-motivated. Also, during the rebuttal, the authors have addressed the reviewers' major concerns by adding comprehensive ablation studies, hyperparameter and complexity analyses, and stronger baselines including full RecVAE and HVamp. By carefully reviewing the reviewers’ comments and the authors’ responses, my final recommendation for this submission is acceptance.

**Reviewer Concerns:**

By carefully reviewing the authors’ responses, I find that the major concerns of the reviewers have been addressed by the authors. The authors have added comprehensive ablation studies, hyperparameter analyses, and stronger baselines including full RecVAE and HVamp. The authors also clarify the geometric effect of PIA and explain how it facilitates collaboration for users with diverse interests and cold-start users.

**Reviewer Scores:**

The initial reviewer scores are 8/8/4/2. By carefully reviewing the reviewers’ comments and the authors’ responses, I believe that the reviewers would be more inclined to maintain an overall positive attitude toward this work.

---

### Decision · Program_Chairs · 2026-01-26

Accept (Poster)